# Osteocytes regulate senescence of bone and bone marrow

Peng Ding[1†], Chuan Gao[1†], Youshui Gao[1†], Delin Liu[2,3], Hao Li[1], Jun Xu[1], Xiaoyi Chen[4], Yigang Huang[1], Changqing Zhang[1*], Minghao Zheng[2,3*], Junjie Gao[1,5*]

[1]Department of Orthopaedics, Shanghai Sixth People's Hospital Affiliated to Shanghai Jiao Tong University School of Medicine, Shanghai, China; [2]Centre for Orthopaedic Translational Research, Medical School, University of Western Australia, Nedlands, Australia; [3]Perron Institute for Neurological and Translational Science, Nedlands, Australia; [4]Ningbo Institute of Life and Health Industry, University of Chinese Academy of Sciences, Ningbo, China; [5]Institute of Microsurgery on Extremities, Shanghai Sixth People's Hospital Affiliated to Shanghai Jiao Tong University School of Medicine, Shanghai, China

*For correspondence:
zhangcq@sjtu.edu.cn (CZ);
minghao.zheng@uwa.edu.au
(MZ);
colingjj@163.com (JG)

[†]These authors contributed
equally to this work

**Competing interest:** The authors declare that no competing interests exist.

## Abstract

The skeletal system contains a series of sophisticated cellular lineages arising from the mesenchymal stem cells (MSCs) and hematopoietic stem cells (HSCs) that determine the homeostasis of bone and bone marrow. Here, we reasoned that osteocyte may exert a function in regulation of these lineage cell specifications and tissue homeostasis. Using a mouse model of conditional deletion of osteocytes by the expression of diphtheria toxin subunit α in dentin matrix protein 1 (DMP1)-positive osteocytes, we demonstrated that partial ablation of DMP1-positive osteocytes caused severe sarcopenia, osteoporosis, and degenerative kyphosis, leading to shorter lifespan in these animals. Osteocytes reduction altered mesenchymal lineage commitment, resulting in impairment of osteogenesis and induction of osteoclastogensis. Single-cell RNA sequencing further revealed that hematopoietic lineage was mobilized toward myeloid lineage differentiation with expanded myeloid progenitors, neutrophils, and monocytes, while the lymphopoiesis was impaired with reduced B cells in the osteocyte ablation mice. The acquisition of a senescence-associated secretory phenotype (SASP) in both osteogenic and myeloid lineage cells was the underlying cause. Together, we showed that osteocytes play critical roles in regulation of lineage cell specifications in bone and bone marrow through mediation of senescence.

## Editor's evaluation

The work provides a new understanding of the role of osteocytes in regulating other lineage cells in bone, bone marrow, and skeletal muscle. The set of data from the genetic mouse model, bone phenotypic analyses, and scRNA-seq analysis supports the conclusion. This is an important and logically presented study that offers new insight into the biology of osteocytes.

## Introduction

The skeletal system is an elaborate organ mainly containing bone, bone marrow, and other connective tissues, whose function includes movement, support, hematopoiesis, immune responses, and endocrine regulation (*Karsenty and Ferron, 2012*; *Katsnelson, 2010*; *Quarles, 2011*). The skeletal system hosts at least more than 12 types of cell lineages arising from hematopoietic stem cells (HSCs) and mesenchymal stem cells (MSCs) (*Méndez-Ferrer et al., 2010*). During hematopoiesis, HSCs

**eLife digest** A hallmark of aging is the weakening of our muscles and bones, which become more fragile as we get older. These gradual changes can result in a humpback and muscle shrinking among other conditions. At the same time little is known about what role osteocytes – the most abundant type of bone cell – play in the process of bone and muscle aging.

One way to investigate the role of osteocytes in aging is to remove them and observe what happens to nearby cells as they age. To achieve this Ding, Gao, Gao et al. genetically altered mice so that they would carry and activate a gene called DTA in their osteocytes. DTA is a gene derived from the bacterium that causes diphtheria, and when it is activated, it produces a toxin that accumulates in cells, eventually killing them. In the mice line developed by Ding, Gao, Gao et al. DTA slowly killed osteocytes, leading to adult mice lacking most of their osteocyte population that have a normal embryonic development. This is important because the fact that the mice develop normally before birth allowed the team to rule out embryonic defects when looking at their results.

Ding, Gao, Gao et al. found that, without enough osteocytes, the nearby bone and bone marrow cells aged faster than expected. Indeed, the skeleton and muscles of adult mice was severely affected by the loss of osteocytes, leading to fragile bones with lower mass and muscle shrinking. These mice looked old in their young age and died earlier.

At the cellular level, the removal of osteocytes impaired the formation of osteoblasts, the cells that are responsible for making bones. It also led to an increase in the numbers of osteoclasts – the cells that destroy bone tissue to repair it and maintain it – and fat tissue cells. Furthermore, cells in the bone marrow, which go on to make white blood cells, were also affected. The mechanisms through which osteocytes affect the growth of these other cells is yet to be fully understood. However, Ding, Gao, Gao et al. did observe that these cells acquired traits characteristic of aging cells, implying that osteocytes have a role in regulating cellular aging or senescence. Among these senescence traits is the increased production and secretion of molecules that interact with the immune system, a feature known as the 'senescence-associated secretory phenotype'.

Overall, the results of Ding, Gao, Gao et al. suggest that reducing the number of osteocytes in mice leads to faster bone aging and affects the balance of the different cell types required for healthy bone and bone marrow growth. Future research could focus on finding drugs that allow osteocytes to keep performing their role during aging, and thus help maintain bone health. The findings of Ding, Gao, Gao et al. also suggest that osteocytes may be playing a previously underappreciated role in age-related diseases, which warrants further investigation.

give rise to lymphoid and myeloid lineage cells, including B cells, neutrophils, monocytes, as well as osteoclasts. Meanwhile, MSCs differentiate into osteoblastic lineage cells, bone marrow adipocytes, and form fibroconnective tissues. The sophisticated processes of differentiation and interaction of these cell lineages are critical not only to skeletal development, but also to the integrity of hematopoietic, immune, and endocrine systems (*Méndez-Ferrer et al., 2010*; *Le et al., 2018*; *Yu and Scadden, 2016*). During aging, these cell lineage commitments change rigorously and cause imbalance between myeloid–lymphoid hematopoiesis and adipo-osteogenic differentiation (*Chen et al., 2016*; *Sinha et al., 2022*), which lead to the increased myelopoiesis and adipogenesis as opposed to lymphopoiesis and osteogenesis. While the complex communications between these cell lineages have been documented, it is still unclear what determines these cell lineages to survive and how their cell fates are maintained during development and aging. It has been speculated that cellular senescence, characterized by cell proliferation arrest, altered metabolism, and apoptosis resistance (*Gorgoulis et al., 2019*; *Tchkonia et al., 2013*), may be responsible for the regulation of lineage cell fates. However, the precise role in aging and age-related diseases remains unclear.

Osteocytes, as the long-living terminally differentiated cells and the most abundant cells within the bone matrix (*Tresguerres et al., 2020*), play vital roles in maintaining the skeletal homeostasis. Apart from mechanical transduction (*Long, 2011*; *Sato et al., 2020*), osteocytes have been shown to regulate bone formation, bone resorption, bone marrow hematopoiesis (*Asada et al., 2013*; *Azab et al., 2020*; *Fulzele et al., 2013*; *Xiao et al., 2021*), and generate endocrine signals to mediate function of other organs (*Razzaque, 2009*; *Fulzele et al., 2017*; *Cain et al., 2012*). Osteocytes regulate

both the osteoblast and osteoclast activities during bone remodeling (*Delgado-Calle and Bellido, 2022*; *Tresguerres et al., 2020*). Sclerostin, one of the key inhibitors of Wnt signaling pathway, is mainly produced by osteocytes (*Tresguerres et al., 2020*). NO and PGE2 secretion by osteocytes in response to mechanical stimulation have anabolic effects on osteoblasts (*Rochefort et al., 2010*). Receptor activation of nuclear factor-κ B ligand (RANKL), the osteoclast differentiation factor, is mainly produced by osteocytes (*Nakashima et al., 2011*). Osteocytes regulate neutrophil development through secretion of soluble factors like IL19 (*Xiao et al., 2021*) and can also regulate myelopoiesis through Gsα-dependent and -independent pathways (*Fulzele et al., 2013*; *Azab et al., 2020*). In addition, studies have shown that aging is associated with dysfunction of osteocytes. Degeneration of osteocytes lacuna-canalicular network had been observed in older adults (*Busse et al., 2010*) and the aging animal model (*Tiede-Lewis et al., 2017*). Senescent osteocytes and their senescence-associated secretory phenotype (SASP) have been shown to contribute to age-related bone loss (*Farr et al., 2016*; *Kim et al., 2020*). Together, current data suggest that osteocyte is a singling cell that coordinates activities of bone and bone marrow during skeletal aging (*Sfeir et al., 2022*).

Here, we hypothesize that coordination of bone and bone marrow homeostasis requires the presence of functional osteocytes. Reduction of osteocytes and their function may result in the detrimental impact in altering lineage cell fates and specifications in bone marrow. Using a mouse model of conditional deletion of osteocytes by the expression of diphtheria toxin subunit α (DTA) in dentin matrix protein 1 (DMP1)-positive osteocytes, we showed that osteocytes regulated bone and bone marrow lineage cell specification. Ablation of osteocytes in these mice caused impairment of osteogenesis and lymphopoiesis, and increased osteoclastogenesis and mobilization of myelopoiesis toward myeloid lineage differentiation with expanded myeloid progenitors, neutrophils, and monocytes. These have resulted in the induction of accelerated skeletal aging. Mice with osteocyte ablation have severe sarcopenia, osteoporosis, and kyphosis at the early stage of 13 weeks, resulting in shorter lifespan. Together, we demonstrated that osteocytes play a critical role in regulation of the HSC and MSC lineage cell differentiations by mediation of senescence.

## Results

### Mice with fewer osteocytes have severe osteoporosis, kyphosis, sarcopenia, and shorter lifespan

To delineate the role of osteocyte in skeletal tissue development and maturation, we established a mouse model based on DTA-mediated cell knockout using the promoter of DMP1 (*Breitman et al., 1990*). The latter is a protein highly expressed in late-stage osteocytes but has been shown not to be essential for early skeletal development (*Feng et al., 2003*). The results showed that complete ablation of DMP1-positive osteocytes (osteocyte[DMP1]) in *Dmp1[cre] Rosa26[em1Cin(SA-IRES-Loxp-ZsGreen-stop-Loxp-DTA)]* homozygotes (DTA[ho]) caused lethality of mice before birth. This has led us to investigate the impact

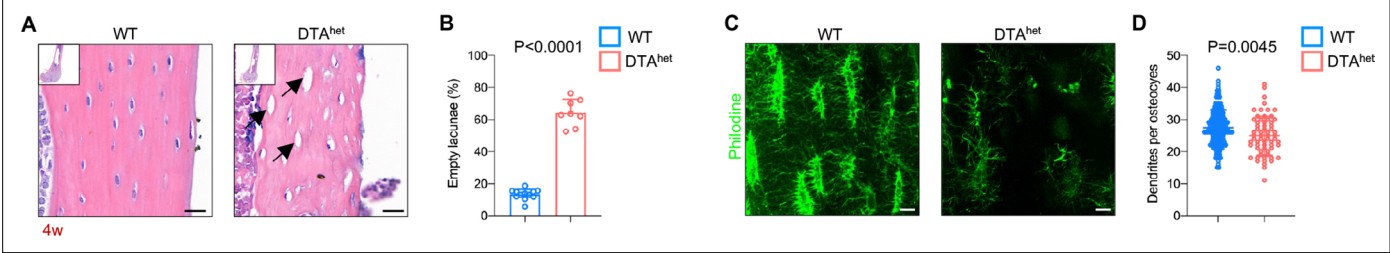

**Figure 1.** DTA[het] mice display partial osteocyte ablation. (**A, B**) Hematoxylin–eosin staining of WT and DTA[het] mice femur at 4 weeks (**A**) and quantification of the ratio of empty lacunae (arrows) (B) (n = 8–12 per group), indicating reduced osteocyte number in DTA[het] mice. Scare bar, 20 μm. (**C, D**) Immunofluorescence staining of femoral cortical bone of 4-week-old WT and DTA[het] mice (**C**) and quantification of dendrites per osteocyte based on the images (**D**) (n = 152 osteocytes in WT group and n = 64 osteocytes in DTA[het] group). Scare bar, 20 μm. Error bar represents the standard deviation.

The online version of this article includes the following source data and figure supplement(s) for figure 1:

**Source data 1.** DTA[het] mice display partial osteocyte ablation.

**Figure supplement 1.** Osteocyte ablation has no impact on embryonic skeletal development.

**Figure supplement 1—source data 1.** DTA[het] mice display partial osteocyte ablation.

of partial ablation of osteocytes using *Dmp1*<sup>cre</sup> *Rosa26*<sup>em1Cin(SA-IRES-Loxp-ZsGreen-stop-Loxp-DTA)</sup> heterozygotes (DTA<sup>het</sup>). Interestingly, Alizarin red/Alcian blue staining of whole-mount skeleton at E19.0 showed no apparent differences of craniofacial, long bones or spines between WT and DTA<sup>het</sup> mice (*Figure 1— figure supplement 1A*). As shown in *Figure 1A and B*, DTA<sup>het</sup> mice had more empty lacunae without the presence of osteocytes within cortical and trabecular bone matrix compared to WT mice. Further, reduced dendrites were also observed in residual osteocytes of DTA<sup>het</sup> mice (*Figure 1C and D*), indicating the impairment of osteocyte network. To define how osteocyte partial ablation was achieved, we performed the quantification of empty lacunae ratio of DTA<sup>het</sup> mice at 13 weeks. About 80% empty lacunae was observed in DTA<sup>het</sup> mice at 13 weeks, which increased by about 20% compared to 4 weeks (*Figure 1—figure supplement 1B and C*), indicating that diphtheria toxin (DT) had an accumulative effect with age in DTA<sup>het</sup> mice. Together, these results indicated that although there was partial ablation of osteocyte<sup>DMP1</sup> in DTA<sup>het</sup> mice, the embryonic development of skeletal tissue appeared to be normal.

Next, we investigated whether reduction of osteocyte<sup>DMP1</sup> in DTA<sup>het</sup> mice had an impact on postnatal maturation of bone tissue. Micro-computed tomography (μCT) examination of the appendicular skeleton revealed a significant decrease in femur bone mineral density (BMD), bone volume fraction (BV/ TV), trabecular number (Tb.N), trabecular thickness (Tb.Th), as well as greater trabecular separation (Tb.Sp) in DTA<sup>het</sup> mice compared to those in WT mice at 4 weeks (*Figure 2A and B*). Moreover, ablation of osteocytes also led to cortical bone loss with decreased cortical thickness (Ct.Th) and increased cortical porosity (Ct.Po) (*Figure 2A and C*). At 13 weeks, DTA<sup>het</sup> mice exhibited more bone loss in both trabecular and cortical bone compared to those in WT mice (*Figure 2D–G*). The progressive bone loss was observed through the life of DTA<sup>het</sup> mice. The phenotype observed is unique and gender-insensitive (*Figure 2—figure supplement 1A–C*). Similarly, μCT observation of axial skeleton also revealed the significant bone loss in vertebral bodies (*Figure 2H and I*, *Figure 2—figure supplement 1D and E*). Furthermore, there was no increase of bone mass of vertebral bodies from 4 to 13 weeks in DTA<sup>het</sup> mice (*Figure 2J and K*), suggesting the retardation of vertebral body maturation. At 13 weeks, obvious kyphosis occurred in DTA<sup>het</sup> mice (*Figure 2L*) due to serve osteoporosis and vertebral body compression. Whole-body μCT scan revealed that there was a giant increase of thoracic and lumbar curvature of DTA<sup>het</sup> mice (*Figure 2M*). At the age of 30 weeks, almost all of DTA<sup>het</sup> mice developed severe kyphosis (*Figure 2N*). Consistent with the development of kyphosis, gait analysis revealed that DTA<sup>het</sup> mice at 4 weeks had abnormal steps when running (*Figure 2—figure supplement 2A and B*). The front and hind stride length were much shorter in DTA<sup>het</sup> mice (*Figure 2—figure supplement 2C*). Also, the swing speed of DTA<sup>het</sup> mice was much slower than WT mice (*Figure 2—figure supplement 2D*, *Videos 1–6*).

Whole-body examination of DTA<sup>het</sup> mice revealed there was a continual body weight loss and muscle weight loss (*Figure 3A–C*) from 4 weeks. Histology examination of gastrocnemius muscles revealed focal muscle atrophy with mild inflammation at 4 weeks (*Figure 3D and E*). Many myonuclei were mispositioned and became centralized in contrast to those in WT mice. No muscle fibrosis was observed. At 13 weeks, there was continual muscle atrophy, rimmed vacuoles, and inclusion bodies seen within the muscle fibers (*Figure 3F and G*). To preclude the direct target of DMP1 on muscle, we quantified the number of muscle fibers and the results showed that there was no reduction of numbers of muscle fibers after osteocyte ablation at 4 weeks (*Figure 3—figure supplement 1A*) and 13 weeks in DTA<sup>het</sup> mice compared to WT mice (*Figure 3—figure supplement 1B*). Measurement of *Dmp1* expression in WT muscle showed that the level of *Dmp1* expression in muscle was very weak and far less than bone (*Figure 3—figure supplement 1C*). Together, these results suggested that DTA<sup>het</sup> mice had systemic muscle atrophy and sarcopenia. It is most likely that sarcopenia was caused by the impairment of osteocyte-muscle crosstalk. Analysis of lifespan in these mice further revealed the average lifespan of DTA<sup>het</sup> mice was about 20–40 weeks, which was much shorter than WT mice (*Figure 3H*). Together, these data demonstrated that osteocytes ablation caused severe osteoporosis and kyphosis, as well as sarcopenia, which occurred at the very early stage compared to normal aging process. These age-related skeletal phenotypes combined with shortened lifespan demonstrated that osteocyte ablation led to the accelerated skeletal aging.

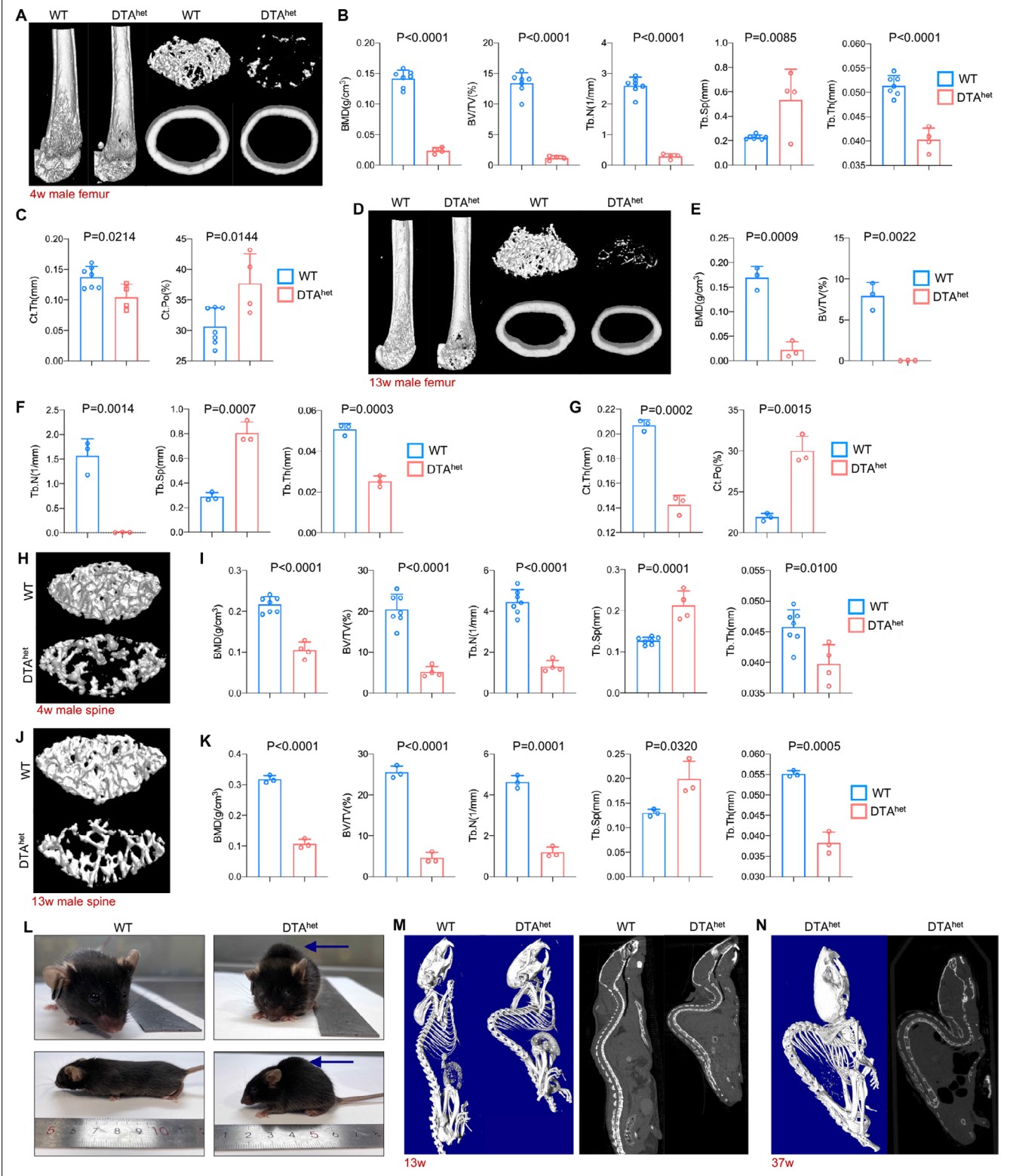

**Figure 2.** Osteocyte ablation induces severe osteoporosis and kyphosis. (**A–C**) Representative micro-computed tomography (µCT) reconstructive images of male WT and DTA^het mice femur at 4 weeks (**A**) and trabecular microstructural parameters (BMD, bone mineral density; BV/TV, bone volume fraction; Tb.N, trabecular number; Tb.Sp, trabecular separation; , Tb.Th, trabecular thickness) (**B**) and cortical microstructural parameters (Ct.Th, cortical thickness; Ct.Po, cortical porosity) (**C**) derived from µCT analysis (n = 4–7 per group). (**D–G**) Representative µCT reconstructive images of male WT and DTA^het mice femur at 13 weeks (**D**) and trabecular microstructural parameters (BMD, BV/TV, Tb.N, Tb.Sp, and Tb.Th) (**E, F**) and cortical microstructural

*Figure 2 continued on next page*

*Figure 2 continued*

parameters (Ct.Th and Ct.Po) (**G**) derived from μCT analysis (n = 3 per group), demonstrating severe bone loss in DTA^het mice. (**H, I**) Representative μCT reconstructive images of male WT and DTA^het mice third lumbar at 4 weeks (**H**) and trabecular microstructural parameters (BMD, BV/TV, Tb.N, Tb.Sp, and Tb.Th) (**I**) derived from μCT analysis (n = 4–7 per group). (**J, K**) Representative μCT reconstructive images of male WT and DTA^het mice third lumbar at 13 weeks (**J**) and trabecular microstructural parameters (BMD, BV/TV, Tb.N, Tb.Sp, and Tb.Th) (**K**) derived from μCT analysis (n = 3 per group), showing vertebral body bone loss in the spine of DTA^het mice. (**L**) Gross images of male WT and DTA^het mice at 13 weeks. (**M**) Representative whole-body μCT reconstructive and sagittal images of male WT and DTA^het mice at 13 weeks. (**N**) Representative whole-body μCT reconstructive and sagittal images of male DTA^het mice at 37 weeks, noting that severe kyphosis occurred in DTA^het mice. Error bar represents the standard deviation.

The online version of this article includes the following source data and figure supplement(s) for figure 2:

**Source data 1.** Osteocyte ablation induces severe osteoporosis in male mice.

**Figure supplement 1.** Osteocyte ablation induces severe osteoporosis and kyphosis.

**Figure supplement 1—source data 1.** Osteocyte ablation induces severe osteoporosis in female mice.

**Figure supplement 2.** Osteocyte ablation induces severe osteoporosis and kyphosis.

**Figure supplement 2—source data 1.** Osteocyte ablation induces abnormal steps of mice.

## Ablation of osteocytes alters mesenchymal lineage commitment and promotes osteoclastogensis

To explore the potential mechanism of why reduction of osteocytes has caused severe osteoporosis and kyphosis, RNA sequencing was performed on whole bone with bone marrow flushed out from DTA^het and WT mice at 4 weeks. Selected skeleton-related Gene Ontology (GO) analysis revealed that downregulated genes by osteocyte ablation were enriched in ossification, osteoblast differentiation, positive regulation of osteoblast differentiation, endochondral ossification, and bone morphogenesis (*Figure 4—figure supplement 1A* and *Supplementary file 1*). Heatmap of significantly differentiated genes (fold change >2.0-fold, WT average FPKM > 10, false discovery rate [FDR] < 0.05) and subsequent RT-qPCR verified that genes that are critical for osteogenesis, including *Alpl*, *Bglap*, *Col1a1*, *Spp1*, *Sp7*, and *Runx2*, were affected by the ablation of osteocytes (*Figure 4—figure supplement 1B and C*). Also, gene set enrichment analysis (GSEA) revealed that osteogenesis-related pathways, including Wnt signaling pathway, Hedgehog signaling pathway, and Notch signaling pathway, were downregulated (*Figure 4—figure supplement 1D–F*). In addition, the number of osteoblasts (N.Ob/BS) and osteoid-covered surface (OS/BS) was remarkably reduced in DTA^het mice compared to WT mice (*Figure 4A and B*). Also, bone marrow fat accumulation in DTA^het mice was observed (*Figure 4C and D*). Together, these results suggested that DTA^het mice displayed increased adipogenesis and decreased osteogenesis. To further evaluate the dynamics of bone formation in DTA^het mice, a 7-day dynamic histomorphometric analysis using calcein labeling was performed. The result showed that mineral surface (MS/BS), mineral apposition rate (MAR), and bone formation rate (BFR/BS) were significantly decreased in DTA^het mice (*Figure 4E and F*). Serum procollagen type 1 N-terminal propeptide (P1NP), a bone formation index, was also reduced after osteocyte ablation (*Figure 4G*). Intriguingly, in vitro osteogenesis showed that there were also decreased osteogenesis and mineralization in DTA^het mice compared to WT mice at both time points of 4 and 13 weeks and the impairment

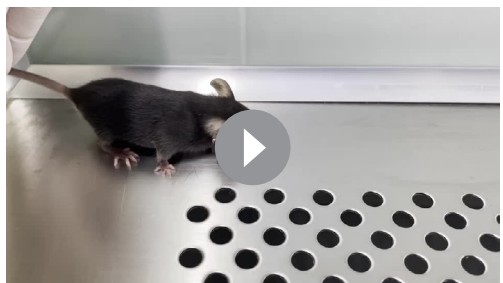

**Video 1.** Representative movie showing movement in WT mice at 4 weeks.

https://elifesciences.org/articles/81480/figures#video1

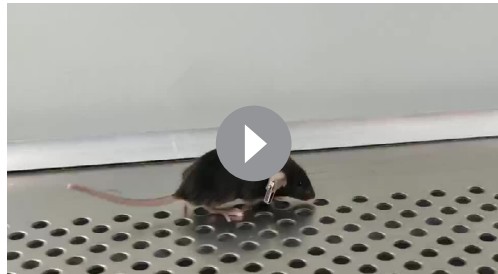

**Video 2.** Representative movie showing movement defects in DTA^het mice at 4 weeks.

https://elifesciences.org/articles/81480/figures#video2

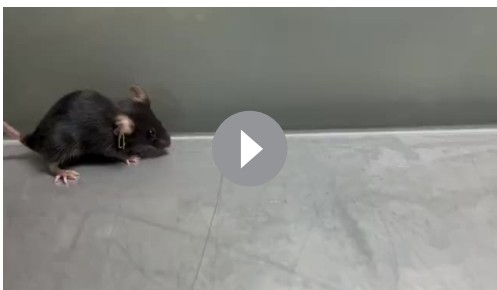

**Video 3.** Representative movie showing movement in WT mice at 13 weeks.
https://elifesciences.org/articles/81480/figures#video3

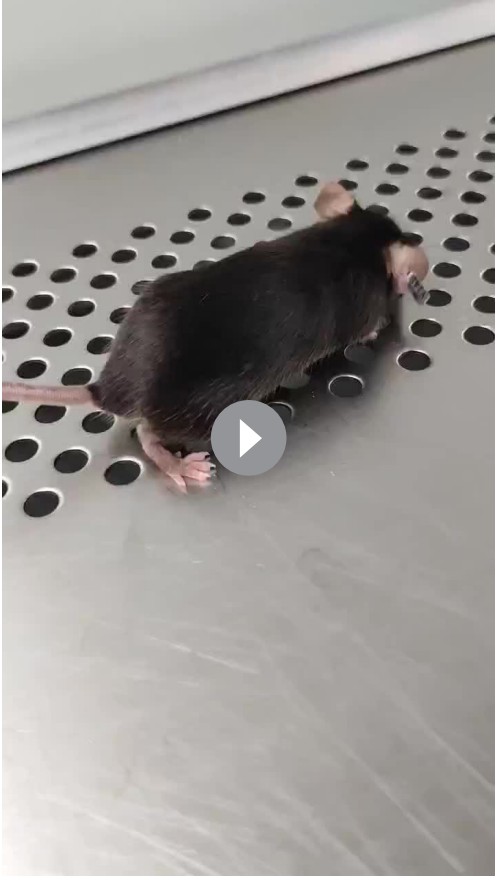

of osteogenesis was greater in DTA[het] mice at 13 weeks compared to 4 weeks (*Figure 4H and I*). And the mRNA level of osteogenic markers at 4 weeks, including *Alpl*, *Bglap*, and *Runx2*, was also decreased (*Figure 4J*).

In the aspect of osteoclastogenesis, histomorphometry analysis revealed that osteoclast surface (Oc.S/BS) and numbers (N.Oc/BS) were significantly increased after osteocytes deletion (*Figure 4K and L*). Circulatory RANKL was also increased in DTA[het] mice (*Figure 4M*). In contrast, circulatory osteoprotegrin (OPG), a decoy receptor of RANKL, was decreased (*Figure 4M*), leading to the elevated ratio of RANKL/OPG (*Figure 4M*). Serum collagen type I c-telopeptide (CTX), a bone resorption index, was also significantly augmented in DTA[het] mice compared to WT mice (*Figure 4N*), which implicated a high level of osteoclast activity of DTA[het] mice in vivo.

**Video 5.** Representative movie showing movement in WT mice at 37 weeks.
https://elifesciences.org/articles/81480/figures#video5

Also, flow cytometry analysis revealed that there was a slight increase (less than 1%) of osteoclast progenitors (B220[-]CD11b[lo]Ly-6C[hi]) in DTA[het] mice at 4 weeks compared to WT mice (*Figure 4—figure supplement 1G and H*). To assess the effects of osteocyte ablation on osteoclastogenesis, bone marrow-derived macrophages (BMMs) from WT and DTA[het] mice at both time points of 4 and 13 weeks were collected and plated at the same density for the examination of osteoclastogenesis in vitro. The results showed that osteoclastogenesis was increased in DTA[het] mice compared to WT mice at both time points (*Figure 4O and Q*). Interestingly, the induction of osteoclastogenesis was greater in DTA[het] mice at 13 weeks compared to 4 weeks (*Figure 4P and Q*), suggesting the time-dependent accumulative effect of osteoclastogenesis in DTA[het] mice. Also, the expression of the signature genes of osteoclasts, including *Acp5*, *Calcr*, and *Ocstamp*, at the mRNA level was significantly upregulated in DTA[het] mice (*Figure 4Q*). Together, osteocytes ablation impaired osteogenesis and promoted osteoclastogenesis.

## Alteration of hematopoietic lineage commitment by osteocyte ablation

As a part of the skeletal system, bone marrow has its vital functions in maintaining bone homeostasis (*Divieti Pajevic and Krause, 2019*; *Fulzele et al., 2013*; *Asada et al., 2013*). HSCs give rise

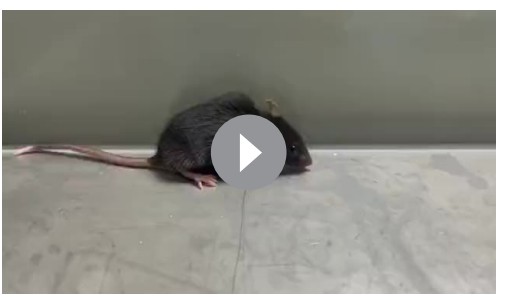

**Video 4.** Representative movie showing movement defects in DTA[het] mice at 13 weeks.
https://elifesciences.org/articles/81480/figures#video4

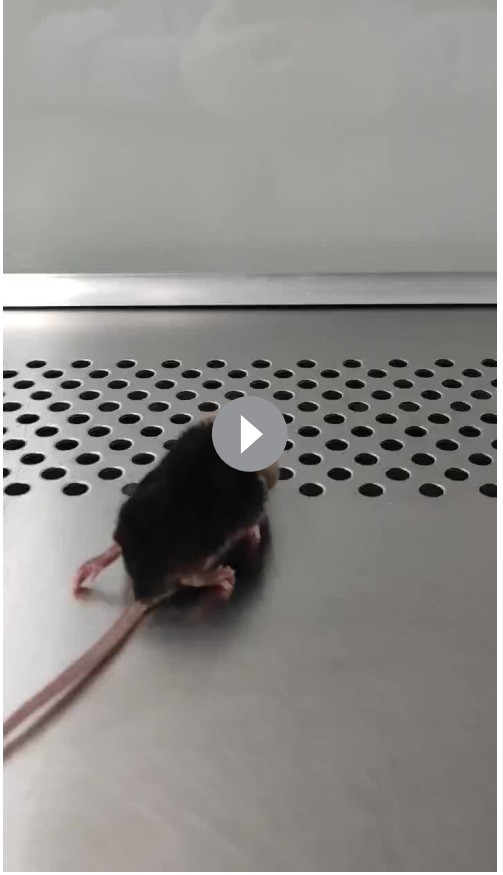

**Video 6.** Representative movie showing movement defects in DTA^het mice at 37 weeks.

https://elifesciences.org/articles/81480/figures#video6

to lymphoid and myeloid lineage cells to establish the hematopoietic and immune system. To gain a full insight into the role of osteocyte in bone marrow homeostasis, single-cell RNA sequencing (scRNA-seq) was performed using 10X Genomics Chromium platform. After rigorous quality control, gene expression data from 26,562 cells (13,835 and 12,727 cells from 4-week littermate WT and DTA^het mice, respectively) were compiled for clustering analysis and revealed 10 distinct populations visualized with Uniform Manifold Approximation and Projection (UMAP) embeddings (*Figure 5A–C*). These 10 distinct populations included B cell, hematopoietic stem cell and progenitor cell (HSPC), megakaryocyte, neutrophil, erythrocyte, monocyte, dendritic cell (DC), macrophage, T cell, and MSC (*Figure 5A and C*). Proportion analysis revealed a significant expansion of neutrophils in DTA^het mice (*Figure 5D and E*). Also, the number of B cells was significantly less in DTA^het mice than that in WT mice (*Figure 5D and E*), which implicated that osteocytes ablation induced lymphoid–myeloid malfunction in the bone marrow. To further dissect the differences in the bone marrow development between the two groups, RNA velocity was performed. The result showed that DTA^het mice had stronger directionality of velocity vectors from the HSPC population to the neutrophil population compared to WT mice (*Figure 5F*), implying that osteocytes deletion altered HSPC differentiation. Meanwhile, myeloid trajectory analysis revealed that there was a significantly higher pseudotime density distribution in G4 cell (a subcluster of neutrophil) in DTA^het mice (*Figure 5G*). In contrast, lymphoid trajectory analysis demonstrated a relatively lower pseudotime density distribution in pre-B cell and immature B cell (subclusters of B cell) in DTA^het mice (*Figure 5H*).

To corroborate the results observed from scRNA-seq, flow cytometry and further analysis were performed after removing adherent cells as previously reported (*Ding et al., 2022*; *Figure 5—figure supplement 1A and B*). Although there was no significant change in HSC (Lin⁻c-Kit⁺Sca1⁺, LSK⁺ cell) numbers between DTA^het and WT mice (*Figure 5—figure supplement 2A and B*), DTA^het mice demonstrated significantly increased number of short-term HSCs (ST-HSCs) with decreased number of long-term HSCs (LT-HSCs), indicating that HSCs in DTA^het mice bone marrow were mobilized (*Figure 5—figure supplement 2C and D*). Further flow cytometry analysis revealed that the number of myeloid progenitors, including common myeloid progenitors (CMP), granulocyte–monocyte progenitors (GMP), and common monocyte progenitors (cMoP), was substantially increased after osteocyte ablation (*Figure 5I and J*, *Figure 5—figure supplement 2E and F*), and megakaryocyte erythroid progenitors (MEP) numbers were decreased (*Figure 5I and J*). Meanwhile, total CD11b⁺ myeloid cells were also increased (*Figure 5K and L*) in DTA^het mice, in which both neutrophils and monocytes significantly expanded (*Figure 5M and N*, *Figure 5—figure supplement 2G and H*). In addition, while the proportion of common lymphoid progenitors (CLP) was not altered in DTA^het mice (*Figure 5—figure supplement 2I and J*), total B220⁺ lymphoid cells reduced remarkably after osteocyte ablation (*Figure 5O and P*), in which DTA^het mice showed a relatively lower proportion of early B cell (pro-B pre-B, immature B, and transitional B cell) and a relatively higher proportion of late B cell (early mature B and late mature B) (*Figure 5Q and R*), which suggested that B cell development

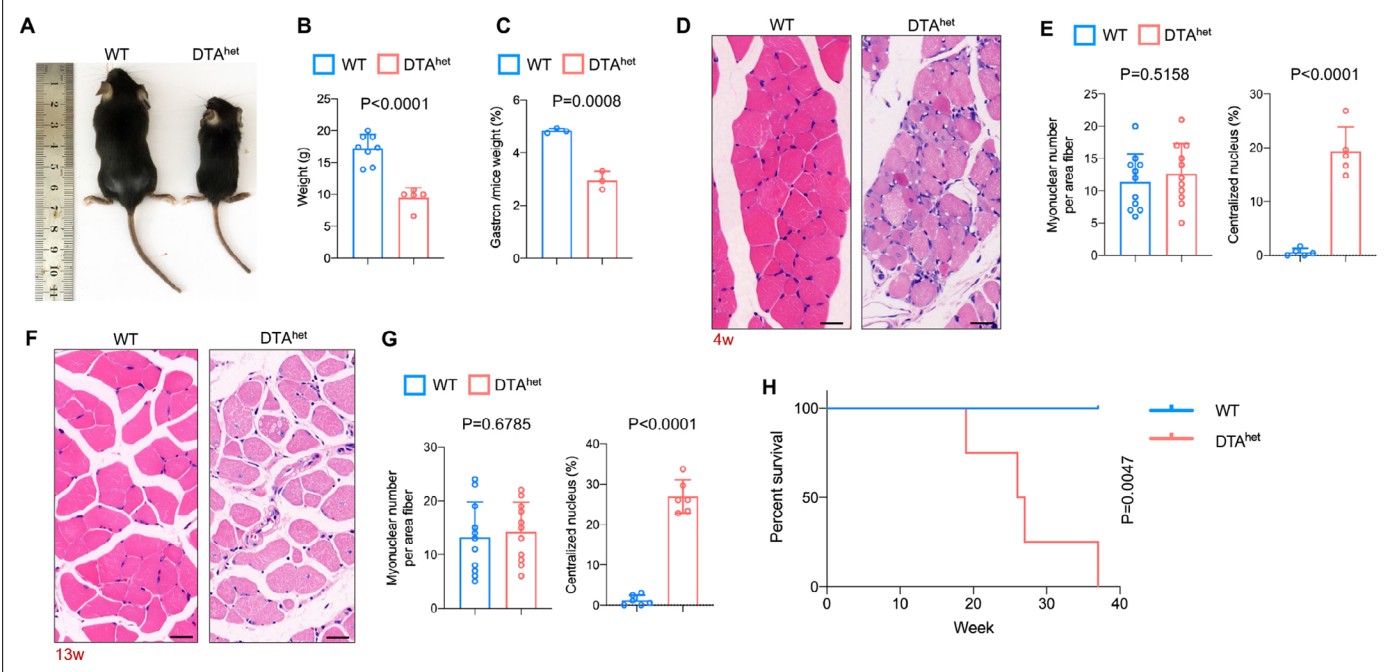

**Figure 3.** Osteocyte ablation leads to severe sarcopenia and shorter lifespan. (**A, B**) Gross images (**A**) and weight (**B**) of male WT and DTA^het mice at 4 weeks (n = 5–8 per group). (**C**) The ratio of gastrocnemius muscle weight in WT and DTA^het mice at 4 weeks (n = 3 per group). (**D, E**) Hematoxylin–eosin staining of WT and DTA^het mice gastrocnemius muscle at 4 weeks (**D**) and quantification of myonuclear number per area fiber (n = 11 per group) and centralized nucleus per field (**E**) (n = 5 per group), showing focal muscle atrophy, increased centralized myonuclei, and mild inflammation in DTA^het mice. Scale bar, 20 μm. (**F, G**) Hematoxylin–eosin staining of WT and DTA^het mice gastrocnemius muscle at 13 weeks (**F**) and quantification of myonuclear number per area fiber (n = 11 per group) and centralized nucleus per field (**G**) (n = 6 per group), noting muscle atrophy, rimmed vacuoles, and inclusion bodies within the muscle fibers in DTA^het mice. Scale bar, 20 μm. (**H**) Kaplan–Meier survival curve of WT and DTA^het mice (n = 4–5 per group), showing that DTA^het mice had shorter lifespan than that of WT mice. Error bar represents the standard deviation.

The online version of this article includes the following source data and figure supplement(s) for figure 3:

**Source data 1.** Osteocyte ablation leads to severe sarcopenia and shorter lifespan.

**Figure supplement 1.** Osteocyte ablation leads to severe sarcopenia and shorter lifespan.

**Figure supplement 1—source data 1.** Osteocyte ablation leads to severe sarcopenia.

was impaired along the immature B to mature B cell transition in DTA^het mice. As scRNA-seq revealed that neutrophils underwent a significant change after osteocyte ablation, neutrophil population were further reclustered into four subclusters from G1 to G4 (*Figure 5—figure supplement 3A and B*) and G4 population was significantly increased in DTA^het mice compared to WT mice (*Figure 5—figure supplement 3C and D*), which implied that osteocyte ablation accelerated neutrophil maturation. Consistent with this observation, neutrophil functions, including activation and chemotaxis, were all upregulated in DTA^het mice (*Figure 5—figure supplement 3E and F*). Genes related to glycolysis and necroptosis were also upregulated (*Figure 5—figure supplement 3G and H*), indicating that osteocyte ablation altered neutrophil functions. Together, these results demonstrated that osteocyte ablation altered hematopoietic lineage, characterized by the shift from lymphopoiesis to myelopoiesis.

## Senescence of osteoprogenitors and myeloid lineage cells leads to the accelerated skeletal aging

Senescence occurred during development as a precise programmed cellular process, contributing to cell fate specification, tissue patterning, and transient structure removal (*Muñoz-Espín and Serrano, 2014*; *Rhinn et al., 2019*). Given that DTA^het mice had accelerated skeletal phenotype of aging, including increased myelopoiesis, osteoporosis, kyphosis, and sarcopenia with shortened lifespan, we hypothesized that osteocyte ablation may be associated with senescence of osteoprogenitors and myeloid lineage cells. ScRNA-seq revealed that total bone marrow had increased senescence with a higher SASP score in DTA^het mice compared to WT mice (*Figure 6A*). DTA^het mice also had increased

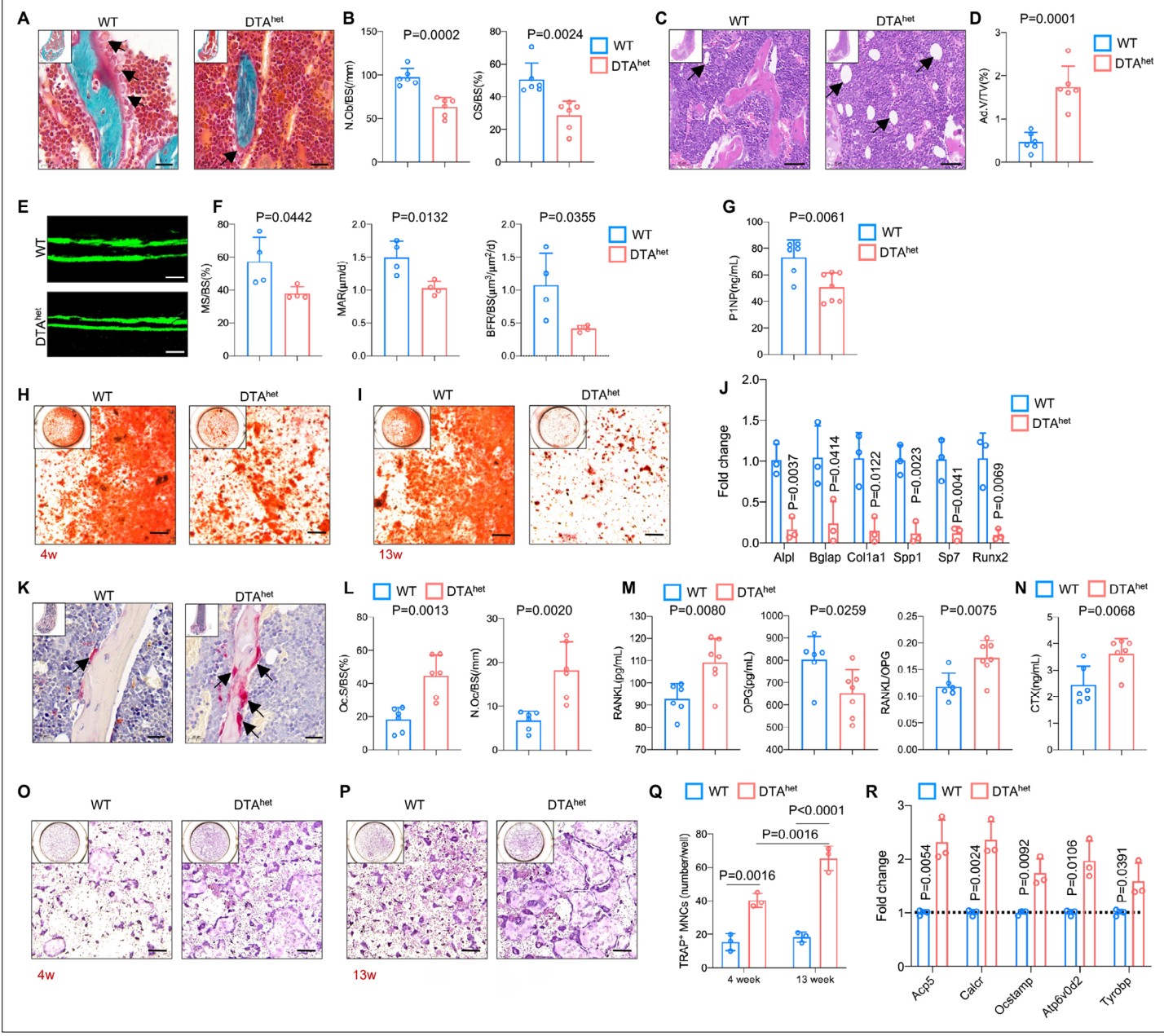

**Figure 4.** Ablation of osteocytes alters mesenchymal lineage commitment and promotes osteoclastogensis. (**A, B**) Goldner trichrome staining of male WT and DTA[het] mice femur at 4 weeks (**A**) and histomorphometry analysis of osteoblast numbers (N.Ob/BS) (arrows) and osteoid-covered surface (OS/BS) (**B**) (n = 6 per group). Scale bar, 20 μm. (**C, D**) Hematoxylin–eosin staining of WT and DTA[het] mice femur at 4 weeks (**C**) and histomorphometry analysis of adipocyte (arrows) volume (Ad.V/TV) (**D**) (n = 6 per group). Scale bar, 50 μm. (**E, F**) Representative images of calcein double labeling of the mineral layers of male WT and DTA[het] mice femur at 4 weeks (**E**) and histomorphometry analysis of the mineral surface (MS/BS), mineral apposition rate (MAR), and bone formation rate (BFR/BS) (**F**) (n = 4 per group). Scale bar, 50 μm. (**G**) ELISA of the concentration of bone formation index P1NP in the serum (n = 6–7 per group). (**H, I**) Alizarin red staining of osteogenesis from 4-week (**H**) and 13-week mice (**I**). Scale bar, 250 μm. (**J**) RT-qPCR analysis of osteoblast signature genes expression at the mRNA levels of osteogenesis from 4-week mice (n = 3 per group from three independent experiments), indicating impaired osteogenesis in DTA[het] mice. (**K, L**) Tartrate-resistant acid phosphatase (TRAP) staining of WT and DTA[het] mice femur at 4 weeks (**K**) and histomorphometry analysis of osteoclast (arrows) surface (Oc.S/BS) and osteoclast numbers (N.Oc/BS) (**L**) (n = 6 per group). Scale bar, 20 μm. (**M**) ELISAs of the concentration of receptor activation of nuclear factor-$\kappa$ B ligand (RANKL), osteoprotegrin (OPG), and the ratio of RANKL/OPG in the serum (n = 6–7 per group). (**N**) ELISA of the concentration of bone resorption index CTX in the serum (n = 6–7 per group). (**O, P**) TRAP staining of osteoclastogenesis from 4-week (**O**) and 13-week mice (**P**) and quantitative analysis (**Q**) of TRAP-positive cells (nucleus > 3) per well (n = 3 per group from three independent experiments). Scale bar, 250 μm. (**R**) RT-qPCR analysis of osteoclast signature genes expression at the mRNA level of

*Figure 4 continued on next page*

Figure 4 continued

osteoclastogenesis from 4-week mice (n = 3 per group from three independent experiments), showing increased osteoclastogensis in DTA^het mice. Error bar represents the standard deviation.

The online version of this article includes the following source data and figure supplement(s) for figure 4:

**Source data 1.** Ablation of osteocytes alters mesenchymal lineage commitment and promotes osteoclastogensis.

**Figure supplement 1.** Ablation of osteocytes alters mesenchymal lineage commitment and promotes osteoclastogensis.

**Figure supplement 1—source data 1.** Ablation of osteocytes alters mesenchymal lineage commitment and promotes osteoclastogensis.

maturity in bone marrow reflected from RNA velocity (*Figure 6B*). Meanwhile, circulatory SASP index, including TNF-α, IL-1β, and IL-6, were also elevated in DTA^het mice (*Figure 6C*). Further scRNA-seq analysis uncovered that MSC, CMP, monocyte, and its subcluster Ly6c2_monocyte, neutrophil, and its subcluster G2, G3, and G4, had increased SASP scores (*Figure 6D–G*) and higher-level expressions of senescence-related genes in DTA^het mice (*Figure 6H*). RT-qPCR also verified the elevated senescence with increased gene expressions, including *Cdkn2a* and *Cdkn1a* in DTA^het mice (*Figure 6I and J*). Further, senescence-associated β-galactosidase (SA-βGal) staining revealed that there were obvious increased numbers of SA-βGal^+ cell in the primary spongiosa, bone marrow, and cortical bone in DTA^het mice compared to WT mice (*Figure 6—figure supplement 1A*). Together, these results suggested that cell senescence of osteoprogenitors and myeloid lineage cells was associated with ablation of osteocyte.

Owning to the fact that osteoblast derives from MSC lineage, we next investigated whether accumulation of osteoprogenitor cell senescence impaired osteogenesis. GO analysis revealed that downregulated genes after osteocyte ablation were enriched in ossification and biomineral tissue development (*Figure 6—figure supplement 1B*), which was consistent with the finding of impaired osteoblast differentiation (*Figure 4H–J*). Meanwhile, the mRNA level of adipogenic markers, including *Adipoq*, *Fabp4*, *Pparg,* and *Cebpa,* was significantly increased (*Figure 6—figure supplement 1C*), indicating increased adipogenesis and alteration of MSC commitment after osteocyte ablation. In addition, the mRNA levels of cartilage anabolism-related genes (*Col1a2*, *Acan*, *Sox9,* and *Prg4*) and catabolism-related genes (*Mmp3*, *Mmp13*, *Adamts1,* and *Adamts5*) were not significantly changed (*Figure 6—figure supplement 1D*), indicating that chondrogenesis was not altered after osteocyte ablation. Similarly, Kyoto Encyclopedia of Genes and Genomes (KEGG) analysis revealed that the subclusters 2 and 4 of Ly6c2^+ monocytes demonstrated the enrichment of osteoclast differentiation-related genes after osteocyte ablation (*Figure 6—figure supplement 1E and F*), which was corroborated in our enhanced in vitro osteoclast differentiation (*Figure 4O–R*). Together, our data suggested that senescence in osteoprogenitors and myeloid lineage cells led to the impaired osteogenesis and increased osteoclastogenesis, respectively.

## Discussion

In this study, we showed that coordination of bone and bone marrow homeostasis requires the presence of functional osteocytes. Reduction of osteocytes number results in the detrimental impact of lineage cell fate and specifications in bone and bone marrow. Partial ablation of osteocytes^DMP1 caused severe sarcopenia, osteoporosis, and degenerative kyphosis, which led to shorter lifespan. Acquisition of SASP in both osteogenic and myeloid lineage cells may be an underlying cause that led to the accelerated skeletal aging phenotype of impaired osteogenesis, increased osteoclastogenesis, and myelopoiesis.

Sarcopenia usually occurs concurrently with osteoporosis during aging (*Clynes et al., 2021*). Our study has shown for the first time that osteocyte ablation caused severe sarcopenia and muscle atrophy. Consistent with our observation, previous studies have reported that osteocyte-specific ablation of Cx43 impaired muscle formation (*Shen et al., 2015*). Osteocyte-derived factors have also been shown to stimulate myogenic differentiation in vitro (*Huang et al., 2017*). On the contrary, specific deletion of Mbtps1 in osteocyte promotes soleus muscle regeneration and increase its size with age (*Gorski et al., 2016*). Sclerostin, an osteocyte-derived circulating protein, is negatively correlated with skeletal muscle mass (*Kim et al., 2019*). Previously there has been a study showing weak *Dmp1* expression in skeletal muscle fibers (*Lim et al., 2017*). This has led us to suggest that sarcopenia may

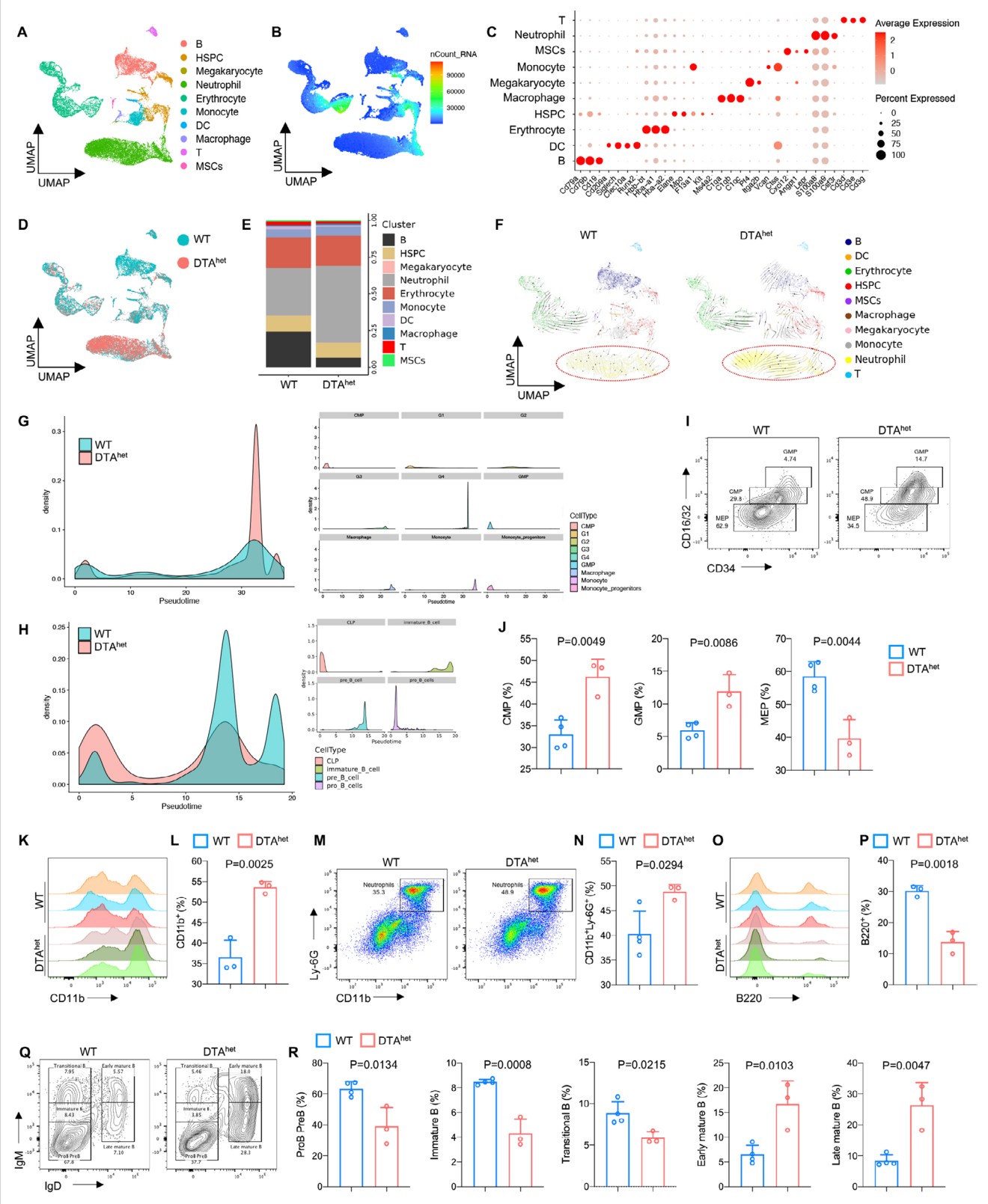

**Figure 5.** Alteration of hematopoietic lineage commitment by osteocyte ablation. (**A, B**) The Uniform Manifold Approximation and Projection (UMAP) plot of cells isolated from the bone marrow of 4-week WT and DTA[het] mice and inferred cluster identity (**A**) and number of mRNA per cell (**B**). (**C**) Dot plot showing the scaled expression of selected signature genes for each cluster. Dot size represents the percentage of cells in each cluster with more than one read of the corresponding gene, and dots are colored by the average expression of each gene in each cluster. (**D, E**) The UMAP plot of cells

*Figure 5 continued on next page*

*Figure 5 continued*

shown by sample (**D**) and proportions of each cluster in two samples (**E**). (**F**) RNA velocity analysis of clusters of WT and DTA^het^ mice shown by the UMAP embedding, showing stronger directionality of velocity vectors from hematopoietic stem cell and progenitor cell (HSPC) cluster to neutrophil cluster in DTA^het^ mice. (**G**) Trajectory analysis of myeloid clusters of WT and DTA^het^ mice, demonstrating myeloid-biased hematopoiesis in DTA^het^ mice. (**H**) Trajectory analysis of lymphoid clusters of WT and DTA^het^ mice, demonstrating impaired lymphopoiesis in DTA^het^ mice. (**I, J**) Representative image of flow cytometry (**I**) and analysis of proportions of myeloid progenitors (common myeloid progenitors [CMP], granulocyte–monocyte progenitors [GMP], and megakaryocyte erythroid progenitors [MEP]) (**J**) of 4-week WT and DTA^het^ mice (n = 3–4 per group). (**K, L**) Representative image of flow cytometry (**K**) and analysis of proportions of CD11b^+^ myeloid cells (**L**) of 4-week WT and DTA^het^ mice (n = 3 per group). (**M, N**) Representative image of flow cytometry (**M**) and analysis of proportions of neutrophils (**N**) of 4-week WT and DTA^het^ mice (n = 3–4 per group). (**O, P**) Representative image of flow cytometry (**O**) and analysis of proportions of B220^+^ lymphoid cells (**P**) of 4-week WT and DTA^het^ mice (n = 3 per group). (**Q, R**) Representative image of flow cytometry (**Q**) and analysis of proportions of ProB PreB, immature B, transitional B, early mature B, and late mature B (**R**) of 4-week WT and DTA^het^ mice (n = 3–4 per group), indicating altered B cell development pattern in DTA^het^ mice. Error bar represents the standard deviation.

The online version of this article includes the following source data and figure supplement(s) for figure 5:

**Source data 1.** Alteration of hematopoietic lineage commitment by osteocyte ablation.

**Figure supplement 1.** Flow cytometry gating strategy.

**Figure supplement 2.** Alteration of hematopoietic lineage commitment by osteocyte ablation.

**Figure supplement 2—source data 1.** Alteration of hematopoietic lineage commitment by osteocyte ablation.

**Figure supplement 3.** Increased granulopoiesis after osteocyte ablation.

be caused directly by the *Dmp1* expression in muscle. However, our histology finding of no obvious changes in the total number of nuclei of muscle in partial ablation of DMP1-positive osteocytes suggested that the sarcopenia and muscle atrophy phenotype is most likely caused by the disturbance of osteocyte-muscle crosstalk. Certainly, further studies based on a more specific osteocyte ablation model are needed to understand the link of osteocytes between osteoporosis and sarcopenia. Nevertheless, severe kyphosis observed in these osteocyte ablation mice supports our hypothesis of direct osteocyte-muscle crosstalk as kyphosis is the direct result of the significant bone loss and sarcopenia (*Wijshake et al., 2012*; *Woods et al., 2020*).

Osteocytes regulate the process of bone resorption and coupled bone formation via secreting factors, including sclerostin and RANKL (*Tresguerres et al., 2020*; *van Bezooijen et al., 2005*; *Nakashima et al., 2011*). Theoretically, osteocyte ablation may lead to lower expression of sclerostin and RNAKL, which in term increased osteogenesis and impaired osteoclastogenesis. However, our results demonstrated that osteocyte ablation impaired osteogenesis and induced osteoclastogenesis. In mice with partial ablation of osteocytes, expression of sclerostin was reduced but the serum RNAKL was increased. In addition, osteogenesis-related pathways, including Wnt signaling pathway, Hedgehog signaling pathway, and Notch signaling pathway, were also downregulated. We speculated that induction of SASP in both osteoprogenitors and myeloid progenitors may account for the underlying cause. Senescent osteoprogenitors have reduced self-renewal capacity and predominantly differentiate into adipocytes as opposed to osteoblasts (*Chen et al., 2016*; *Li et al., 2017*; *Rosen et al., 2009*). Consistently, our model indicated an increased adipogenesis after osteocyte ablation. Also, fat-induction factors inhibit osteogenesis during adipogenesis (*Chen et al., 2016*). Thus, senescence accumulation in osteoprogenitors led to the impaired osteogenesis. As for enhanced osteoclastogenesis, besides the production of RANKL from osteogenic cells like osteocytes and osteoblasts (*Nakashima et al., 2011*; *Fumoto et al., 2014*), other cells like adipocyte and T cell, also secret RANKL to regulate bone metabolism (*Yu et al., 2021*; *Hu et al., 2021*; *Djaafar et al., 2010*; *Takayanagi et al., 2000*). Also, B cell can produce OPG to regulate RANKL/OPG axis (*Li et al., 2007*). In our model, increased adipogenesis, T cell expansion (data not shown), and decreased B cell number may compensate for the altered RANKL/OPG axis. Intriguingly, we also found that even under in vitro condition in which osteocyte ablation no longer exists, impairment of osteogenesis and induction of osteoclastogenesis were still observed. Our study has suggested that osteoprogenitors and BMMs have been primed by the altered bone microenvironment in DTA^het^ mice before in vitro differentiation. In support of this, previous studies have suggested that progenitor cells can receive a long-lasting impact from the in vivo local microenvironment, where these cells are situated. Isolation of cells for in vitro cell differentiation or even transferring cells to healthy mice would not alter their original in vivo phenotypes (*Cao et al., 2020*; *Isaac et al., 2014*; *Ding et al., 2022*; *Edgar et al., 2021*; *Li et al., 2022*).

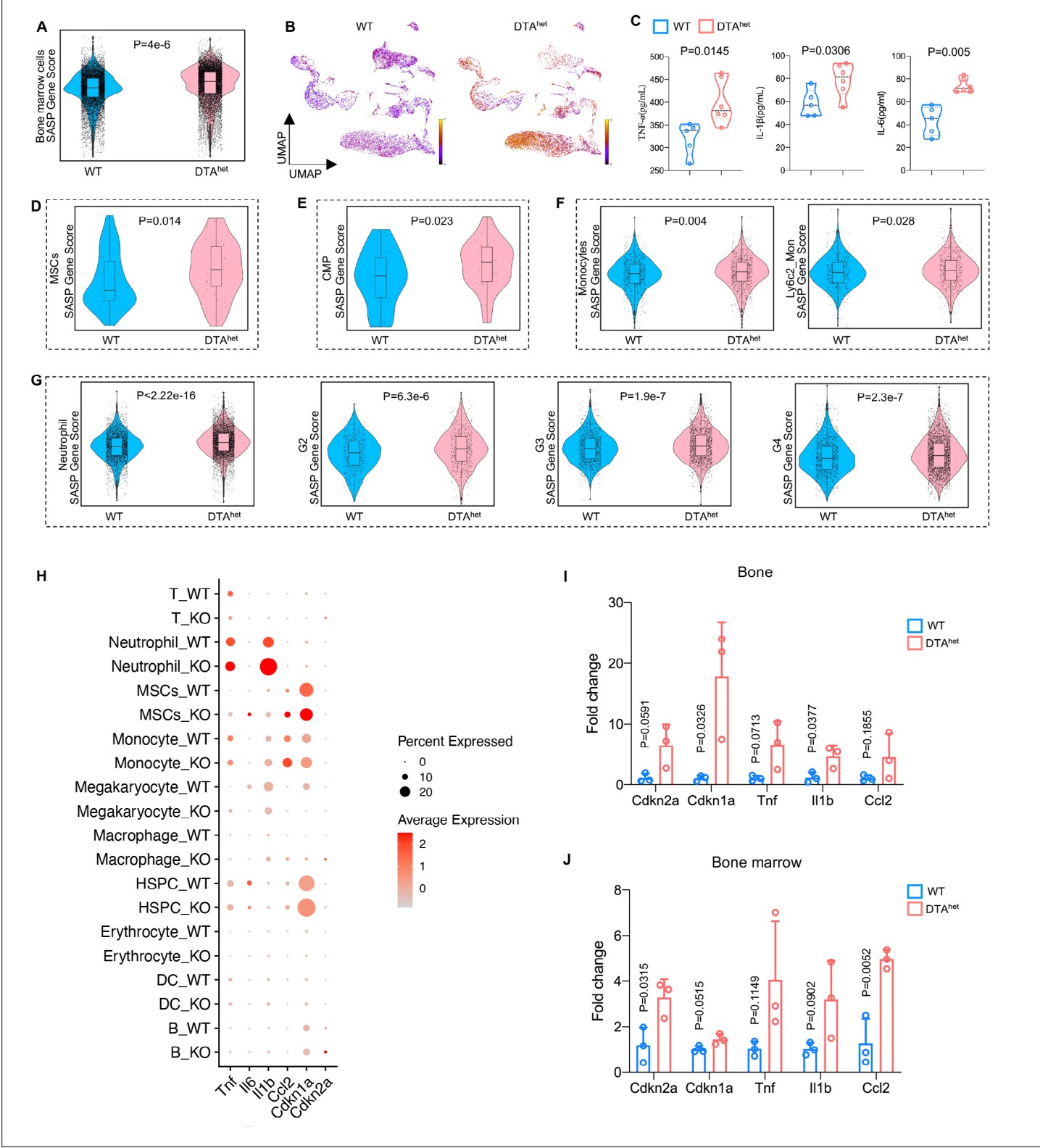

**Figure 6.** Senescence of osteoprogenitors and myeloid lineage cells leads to the accelerated skeletal aging. (**A**) Comparisons of total bone marrow cells senescence-associated secretory phenotype (SASP) score between 4-week WT and DTA^het^ mice. (**B**) Latent time of RNA velocity analysis of WT and DTA^het^ mice shown by the Uniform Manifold Approximation and Projection (UMAP) embedding. (**C**) ELISAs of the concentration of TNF-α, IL-1β, and IL-6 of 4-week WT and DTA^het^ mice in the serum (n = 5–6 per group). (**D**) Comparisons of mesenchymal stem cells (MSCs) SASP score between

*Figure 6 continued on next page*

*Figure 6 continued*

4-week WT and DTA^het mice, indicating the senescence of osteoprogenitors in DTA^het mice. (**E**) Comparisons of common myeloid progenitors (CMP) SASP score between 4-week WT and DTA^het mice. (**F**) Comparisons of monocytes and its subcluster Ly6c2_monocytes SASP score between 4-week WT and DTA^het mice. (**G**) Comparisons of neutrophils and its subcluster (G2, G3, and G4) SASP score between 4-week WT and DTA^het mice, indicating the senescence of myeloid lineage cells. (**H**) Bubble plot of the expression of senescence-related genes in subclusters of WT and DTA^het mice. (**I**) RT-qPCR analysis of senescence-related genes expression at the mRNA level of 4-week WT and DTA^het mice cortical bone (n = 3 per group). (**J**) RT-qPCR analysis of senescence-related genes expression at the mRNA level of 4-week WT and DTA^het mice bone marrow (n = 3 per group). Error bar represents the standard deviation.

The online version of this article includes the following source data and figure supplement(s) for figure 6:

**Source data 1.** Senescence of osteoprogenitors and myeloid lineage cells leads to the accelerated skeletal aging.

**Figure supplement 1.** Senescence of osteoprogenitors and myeloid lineage cells leads to the accelerated skeletal aging.

**Figure supplement 1—source data 1.** Osteocyte ablation induced MSC lineage towards adipogenesis.

Bone marrow, embedded in the skeletal system, has a close link with matrix-embedded osteocyte. Previous studies have reported that osteocyte regulates myelopoiesis via Gsα-dependent and -independent signaling (*Fulzele et al., 2013*; *Azab et al., 2020*). A recent study also reported that osteocyte mTORC1 signaling regulates granulopoiesis via secreted IL-19 (*Xiao et al., 2021*). Meanwhile, sclerostin secreted by osteocyte adversely affects B cell survival (*Horowitz and Fretz, 2012*). In our study, when osteocytes were partially depleted, myelopoiesis, especially granulopoiesis, was significantly induced, but B cell development was significantly impaired. Further studies demonstrated that HSCs were mobilized and shifted to myelopoiesis with increased CMP, GMP, cMoP, and CD11b⁺ myeloid cells, in which monocytes and neutrophils were increased, and neutrophil function was also activated after osteocyte ablation. While B cell number was severely reduced with altered development pattern. Interestingly, a previous study has shown that osteoblastic cell supports megakaryopoiesis and platelet formation (*Xiao et al., 2017*). In our study, the number of MEP (erythrocyte and platelet precursors) was also reduced, and scRNA-seq analysis showed no significant change in erythrocyte population (data not shown), inferring that osteocyte may also participate in regulating platelet formation.

Bone and bone marrow harbor different cell lineages and form specific niches to maintain complex, delicate and extensive communications between them (*Hu et al., 2016*). Previous studies have shown that osteocyte controls bone remodeling, regulates hematopoiesis, and even remote organ function (*Divieti Pajevic and Krause, 2019*; *Asada et al., 2015*), via secretion of factors, including sclerostin, RANKL, FGF23, and IL19 (*Xiao et al., 2021*). Although our study has shown that osteocytes also influence cell lineage commitments of bone and bone marrow via the mediation of cell senescence, it is still not clear what factors osteocytes produce to regulate this process. Further study is required to identify the mechanism in which osteocytes regulate the homeostasis of bone and bone marrow. Furthermore, as our study only focuses on the effect of osteocyte ablation in muscle, bone, and bone marrow, it is still not clear what is the impact of osteocyte ablation on other organs. Nevertheless, a previous study showed that osteocyte ablation induces lymphoid organ atrophy, thymocyte depletion, and altered fat metabolism in 'osteocyte-less' mice model (*Sato et al., 2013*), suggesting the role of osteocytes in the extraskeletal system.

In conclusion, we demonstrated a critical role of osteocytes in regulating senescence of bone and bone marrow (*Figure 7*). Ablation of osteocytes induced SASP accumulation in bone marrow osteoprogenitors and myeloid lineage cells, which altered MSC and HSC lineage commitments with impaired osteogenesis, and promoted myelopoiesis and osteoclastogenesis, leading to the accelerated skeletal aging phenotype with severe sarcopenia, osteoporosis, degenerative kyphosis, and bone marrow myelopoiesis, thus shortened lifespan of mice. Targeting osteocyte function and cell fate may shed light on the therapeutic regimens for aging-associated bone diseases.

## Materials and methods
### Mice
All mouse lines were maintained on a C57BL/6J background. *Dmp1*^cre mice were provided by J. Q. (Jerry) Feng from Texas A&M College of Dentistry, USA (Jackson Laboratory stock number: 023047).

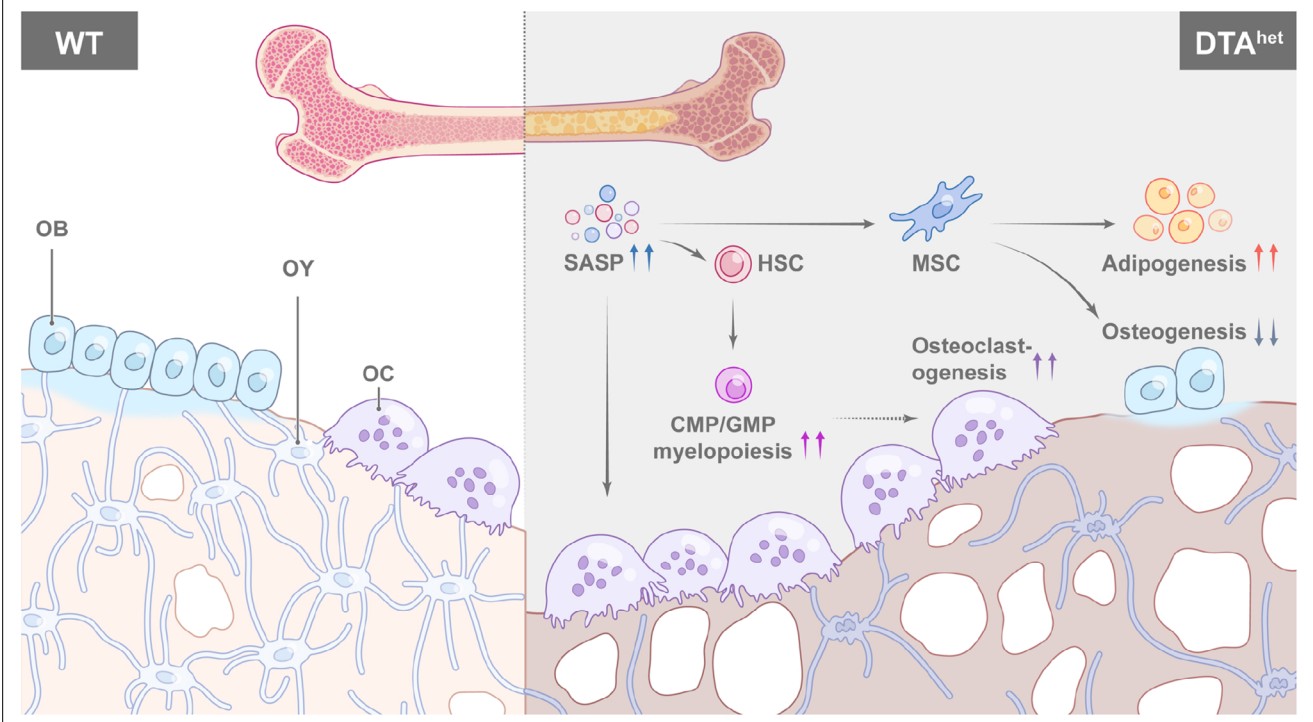

**Figure 7.** Schematic diagram of osteocyte ablation-induced skeletal senescence. Ablation of osteocytes induced senescence-associated secretory phenotype (SASP) accumulation in bone marrow osteoprogenitors and myeloid lineage cells, which altered mesenchymal stem cell (MSC) and hematopoietic stem cell (HSC) lineage commitments with promoted adipogenesis, myelopoiesis, and osteoclastogenesis at the expense of osteogenesis and lymphopoiesis, leading to the accelerated skeletal aging phenotype with severe sarcopenia, osteoporosis, degenerative kyphosis, and bone marrow myelopoiesis, thus shortened lifespan of mice.

*Rosa26*<sup>em1Cin(SA-IRES-Loxp-ZsGreen-stop-Loxp-DTA)</sup> heterozygotes were from GemPharmatech (strain ID: T009408). Osteocyte ablation mice model during development was established by crossing *Dmp1*<sup>cre</sup> mice with *Rosa26*<sup>em1Cin(SA-IRES-Loxp-ZsGreen-stop-Loxp-DTA)</sup> homozygotes to obtain *Dmp1*<sup>cre</sup> *Rosa26*<sup>em1Cin(SA-IRES-Loxp-ZsGreen-stop-Loxp-DTA)</sup> heterozygotes (DTA<sup>het</sup>). All mice experiments were approved by the Animal Care and Use Committee of Shanghai Sixth People's Hospital (permit number: 2021-0935, 2021-0936). All surgeries were performed under anesthesia using isoflurane or sodium pentobarbital, and every effort was made to minimize suffering.

## Bone histomorphometry analysis

Mice femur was dissected and fixed in 4% paraformaldehyde (PFA) for 2 days and further decalcified with 10% EDTA (pH = 7.2) at 4°C for about 2 weeks. Then, specimens were embedded in paraffin and sectioned at 4 μm thickness. TRAP staining was performed for osteoclast analysis. Hematoxylin–eosin (H&E) staining was performed for adipocyte and osteocyte analysis. For osteoblast analysis, undecalcified femur was embedded in plastic and sectioned at 5 μm thickness and Goldner trichrome staining was performed. For dynamic histomorphometry analysis, double calcein labeling was used. Briefly, each mouse was given 30 μg/g body weight calcein (Sigma) on days 1 and 7 by intraperitoneal injection before sacrifice. Bones were then fixed, dehydrated, embedded in plastic, cut into 5 μm slices, and calculated using the software under fluorescence. BioQuant Osteo software (BioQuant) was used for histomorphometry analysis. Accepted nomenclature was used to report the results (*Dempster et al., 2013*). ImageJ was used to measure the number of osteocyte lacunae.

## Immunofluorescence staining

Both ends of the mice's tibias/femurs were removed. Then, they were embedded in OCT for frozen sectioning and cut parallel to the long axis of the long bones. Stop cutting when the maximum cross section of the long bones was observed. The OCT around the rest of the bones were melted at room

temperature. The remaining bone samples were washed three times in PBS for 10 min and fixed in 4% PFA for 2 hr. Then, they were immersed in 0.1% Triton X-100 for 1 hr, blocked using 3% BSA, and stained using Alexa Fluor 568 Phalloidin (Invitrogen) for 48 hr at 4°C in the dark with gentle shake. The samples were washed three times with PBS for 10 min. The cross section of the sample was inverted in the confocal dish. Pictures were captured using confocal microscopy (Olympus), and ImageJ was used to measure the number of dendrites per osteocyte.

## SA-βGal staining

For SA-βGal staining, mice femur was dissected and fixed in 4% PFA for 2 days and further decalcified with 10% EDTA (pH = 7.2) at 4°C for about 2 weeks. Then, specimens were dehydrated in 30% sucrose and embedded in OCT and frozen sectioned at 10 μm thickness. Then, SA-βGal staining was performed according to the manufacturer's instructions (Beyotime).

## Bone density measurements

Mice femurs and L3 lumbar were stripped of soft tissue and fixed in 4% PFA overnight at 4°C, then stored in 70% ethanol until scanned using the μCT instrument (SkyScan 1176). Relevant structure parameters of the μCT instrument were as previously reported (*Ding et al., 2022*): scanning voxel size, $9 \times 9 \times 9$ μm$^3$; X-ray tube potential, 50 kV and 450 μA; integration time, 520 ms; and rotation step, 0.4° for 180° scanning. CTAn micro-CT software version 1.13 (Bruker) was used to analyze the images. The threshold value (grayscale index) for all trabecular bone was 75. For all cortical bones, the threshold value (grayscale index) was 110. The femurs were analyzed at a resolution of 9 μm. The volumetric regions for trabecular analyses include the secondary spongiosa located 1 mm from the growth plate and extending 1.8 mm (200 sections) proximally. For cortical bone analysis, the volumetric regions include 600 μm long at mid-diaphysis of the femur (300 μm extending proximally and distally from the diaphyseal midpoint between the proximal and distal growth plates). For vertebrae, the volumetric regions include the entire trabecular region without the primary spongiosa (300 μm below the cranial and above the caudal growth plate). Morphometric parameters, including BMD, BV/TV, Tb.N, Tb.Th, Tb.Sp, Ct.Th, and Ct.Po, were calculated.

## Gait analysis

CatWalk automated gait analysis system (Noldus Information Technology) was used to analyze gait. Mice were expected to run along a special glass plate with a green LED lit and a high-speed video camera under it. Their paws were captured by the camera. Before the formal experiments, the mice were habituated in the plate to achieve an unforced locomotion. Three compliant runs without stopping, changing direction, and turning around were analyzed with CatWalk Software. Relevant data were generated by CatWalk Software after each footprint was checked manually. Data including stride length, swing speed, and normal step sequence radio were analyzed.

## Whole-mount Alcian blue/Alizarin red staining

The skin and viscera of the intact fetal mice (E19.0) were removed. The embryos were fixed in 95% ethanol overnight and then degreased in absolute acetone overnight with gentle agitation. The embryos were stained overnight in 0.015% Alcian blue (Sigma)/0.005% Alizarin red (Sigma) in 70% ethanol with gentle agitation. They were washed in 70% ethanol for 30 min three times and digested using 1% KOH solution. When most of the soft tissue was digested, the embryos were immersed in 75% (vol/vol)/1% KOH/glycerol solution for further clearing. Graded glycerol was changed according to the degree of embryos digestion and relevant pictures were obtained under the microscope (Leica).

## Whole-body μCT scan

The 13- and 37-week-old DTA[het] and WT mice were deeply anesthetized and carefully positioned with a dedicated cradle and holder to capture the whole-body (excluding the tail) radiographs at a resolution of 35 μm using the μCT instrument (SkyScan 1176). Scanning details are listed as follows: X-ray tube potential, 65 kV and 375 μA; exposure time, 150 ms; and rotation step, 0.5° for 180° scanning. CTAn micro-CT software version 1.13 (Bruker) was used to reconstruct pictures.

## RNA-seq

Total RNA of whole bone with bone marrow flushed out from 4-week WT and DTA[het] mice was extracted using TRIzol reagent (Thermo Fisher), quantified and purified using Bioanalyzer 2100 and RNA 6000 Nano LabChip Kit (Agilent). Following purification, mRNA library was constructed, fragmented, amplified, and loaded into the nanoarray and sequencing was performed on Illumina NovaSeq 6000 platform following the vendor's recommended protocol. After sequencing, generated reads were filtered and mapped to the reference genome using HISAT2 (v2.0.4) and assembled using StringTie (v1.3.4d) with default parameters. Then, all transcriptomes from all samples were merged to reconstruct a comprehensive transcriptome using GffCompare software (v0.9.8), and the expression levels of all transcripts were calculated by Stringtie and ballgown. Differential gene analysis was performed by DESeq2 software and then subjected to enrichment analysis of GO functions. GSEA was performed using GSEA software (version 4.1.0; Broad Institute, MIT). Genes were ranked according to their expression; gene sets were searched from website (https://www.gsea-msigdb.org). The data were deposited into the GEO repository (GSE202356).

## Cell culture

### In vitro osteoclastogenesis assay

The bone marrow of mice femurs and tibias were flushed to get bone marrow cells. Cells were cultured overnight by using α-MEM (Hyclone), which contains 10% FBS (Gibco), 100 µg/ml streptomycin (Gibco), and 100 U/ml penicillin (Gibco). The nonadherent cells were collected, layered on Ficoll-Paque (GE Healthcare), and separated through density gradient centrifugation at 4°C and 2000 rpm for 20 min. The BMMs in the middle layer of the separation were collected and washed twice with ice-cold PBS. To induce osteoclast differentiation, BMMs ($2.5 \times 10^4$ cells per well for 96-well plates and $8 \times 10^5$ per well for 6-well plates) were cultured by using α-MEM, which contains 10% FBS, 100 µg/ml streptomycin, 100 U/ml penicillin, 100 ng/ml M-CSF (PeproTech), and 100 ng/ml RANKL (PeproTech) for 5 days before TRAP staining. Cells were cultured at 37°C in a humidified incubator at 5% $CO_2$. The medium was changed every 2 days. At the end of assay (the fifth day), the cells were fixed and stained with tartrate-resistant acid phosphatase (TRAP) kit according to the manufacturer's instructions (Sigma) to quantify osteoclast numbers, or RNA was extracted as per the recommended protocol. TRAP-positive cells that contain more than three nuclei were counted as mature osteoclast-like cells (OCLs). The assay was repeated three times, and the number of OCLs per well was recorded for each biological replicate.

### Isolation of mesenchymal stem cells and trilineage differentiation

Bone marrow cells were collected by flushing femur and tibia from WT and DTA[het] mice and were cultured in DMEM (Hyclone) containing 10% FBS, 100 U/ml penicillin, and 100 µg/ml streptomycin. After 48 hr, nonadherent cells were removed and fresh medium was added. The adherent spindle-shaped cells were further cultured for 2 days. After culturing the cells to 70–80% confluence, they were replated at a density of 5000 cells per well for 96-well plates or $2 \times 10^5$ cells per well for 6-well plates. When the cells were cultured to 70–80% confluence, the medium was replaced with osteogenic differentiation medium (Cyagen) for osteogenesis or with adipogenic differentiation medium (Cyagen) for adipogenesis or with chondrogenic differentiation medium (Cyagen). RNA extraction was performed after 2 days of differentiation. After 3 weeks of differentiation of osteogenesis, Alizarin red staining was performed.

## RT-qPCR

Total RNA was isolated using RNeasy Mini Kit (QIAGEN). 500 ng of total RNA was reverse-transcribed into cDNA using PrimeScript RT Master Mix (Takara, RR036A). qPCR analyses were performed using SYBR Premix Ex Taq Ⅱ (Takara, RR820L) and samples were run on the ABI HT7900 platform (Applied Biosystems). SYBR Green PCR conditions were 1 cycle of 95°C for 30 s, and 40 cycles of 95°C for 5 s and 34°C for 60 s. Melting curve stage was added to check primers' specificity. Relative gene expression levels were calculated using the threshold cycle ($2^{-\Delta\Delta CT}$) method. Relevant primers are listed as follows: *Gapdh*: 5'-ACC CAG AAG ACT GTG GAT GG-3' and 5'-CAC ATT GGG GGT AGG AAC AC-3'; *Cdkn1a*: 5'-GTC AGG CTG GTC TGC CTC CG-3' and 5'-CGG TCC CGT GGA CAG TGA GCA G-3'; *Cdkn2a*: 5'-GTC AGG CTG GTC TGC CTC CG-3' and 5'-CGG TCC CGT GGA CAG TGA GCA G-3';

*Ccl2*: 5′-GCA TCC ACG TGT TGG CTC A-3′ and 5′-CTC CAG CCT ACT CAT TGG GAT CA-3′; *Tnf*: 5′-ATG AGA AGT TCC CAA ATG GC-3′ and 5′-CTC CAC TTG GTG GTT TGC TA-3′; *Il1b*: 5′-GCC CAT CCT CTG TGA CTC AT-3′ and 5′-AGG CCA CAG GTA TTT TGT CG-3′; *Alpl*: 5′-TCA GGG CAA TGA GGT CAC AT-3′ and 5′-CCT CTG GTG GCA TCT CGT TA-3′; *Bglap*: 5′-CCC TGA GTC TGA CAA AGC CT-3′ and 5′-GCG GTC TTC AAG CCA TAC TG-3′; *Col1a1*: 5′-ATA AGT CCC TTC CTG CCC AC-3′ and 5′-TGG GAC ATT TCA GCA TTG CC-3′; *Spp1*: 5′-ATG CCA CAG ATG AGG ACC TC-3′ and 5′-CCT GGC TCT CTT TGG AAT GC-3′; *Sp7*: 5′-TCG GGG AAG AAG AAG CCA AT-3′ and 5′-CAA TAG GAG AGA GCG AGG GG-3′; *Runx2*: 5′-GCC CAG GCG TAT TTC AGA TG-3′ and 5′-GGT AAA GGT GGC TGG GTA GT-3′; *Dmp1*: 5′-CAG TGA GGA TGA GGC AGA CA-3′ and 5′-CGA TCG CTC CTG GTA CTC TC-3′; *Sost*: 5′-GCC GGA CCT ATA CAG GAC AA-3′ and 5′-CAC GTA GCC CAA CAT CAC AC-3′; *Acp5*: 5′-TGG ACA TGA CCA CAA CCT GCA GTA-3′and 5′-TCG CAC AGA GGG ATC CAT GAA GTT-3′; *Calcr*: 5′-AGC CAC AGC CTA TCA GCA CT-3′ and 5′-GAC CCA CAA GAG CCA GGT AA-3′; *Ocstamp*: 5′-TGG GCC TCC ATA TGA CCT CGA GTA G-3′ and 5′-TCA AAG GCT TGT AAA TTG AGG AGT-3′; *Atp6v0d2*: 5′-ACA TGT CCA CTG GAA GCC AGT AA-3′ and 5′-ATG AAC GTA TGA GGC CAG TGA GCA-3′; *Tyrobp*:5′-CTG GTG TAC TGG CTG GGA TT-3′ and 5′-CTG GTC TCT GAC CCT GAA GC-3′; *Adipoq*: 5′-GAC CTG GCC ACT TTC TCC TC-3′ and 5′-TCC TGA GCC CTT TTG GTG TC-3′; *Fabp4*: 5′-GAT GAA ATC ACC GCA GAC GAC A-3′ and 5′-ATT GTG GTC GAC TTT CCA TCC C-3′; *Pparg*: 5′-GGA AAG ACA ACG GAC AAA TCA C-3′ and 5′-TAC GGA TCG AAA CTG GCA C-3′; *Cebpa*: 5′-TGG ACA AGA ACA GCA ACG AG-3′ and 5′-TCA CTG GTC AAC TCC AGC AC-3′; *Col1a2*: 5′-GGG AAT GTC CTC TGC GAT GAC-3′ and 5′-GAA GGG GAT CTC GGG GTT G-3′; *Acan*: 5′-CCT GCT ACT TCA TCG ACC CC-3′ and 5′-AGA TGC TGT TGA CTC GAA CCT-3′; *Sox9*: 5′-CGG AAC AGA CTC ACA TCT CTC C-3′ and 5′-GCT TGC ACG TCG GTT TTG G-3′; *Prg4*: 5′-GGG TGG AAA ATA CTT CCC GTC-3′ and 5′-CAG GAC AGC ACT CCA TGT AGT-3′; *Mmp3*: 5′-ACA TGG AGA CTT TGT CCC TTT TG-3′ and 5′-TTG CTG AGT GGT AGA GTC CC-3′; *Mmp13*: 5′-CTT CTT CTT GTT GAG CTG GAC TC-3′ and 5′-CTG TGG AGG TCA CTG TAG ACT-3′; *Adamts1*: 5′-CAT AAC AAT GCT GCT ATG TGC G-3′ and 5′-TGT CCG CTG CAA CTC AG-3′; *Adamts5*: 5′-GGA GCG AGG CCA TTT ACA AC-3′ and 5′-CGT AGA CAA GGT AGC CCA CTT T-3′. All these primers were synthesized by Sangon Biotech Company (Shanghai).

## Flow cytometry

Bone marrow cells were isolated by flushing the bone marrow of mice femurs and tibias with PBS and were dissociated into a single-cell suspension by gently filtering them through 70 μm nylon mesh. After red blood cells lysis, the isolated cells were blocked by anti-mouse CD16/32 antibody (BioLegend, 101302) for 15 min and stained with fluorescence-conjugated antibodies for 30 min at 4°C in the dark. Relevant antibodies are listed as follows and their catalog numbers are provided in the parentheses: anti-Ly-6C-Pacific Blue (128013), anti-Ly-6C-PE (128007), anti-Ly-6G-Pacific Blue (127611), anti-Ly-6G-PE/Cy7 (127617), anti-CD16/32-FITC (101305), anti-CD115-PE (135505), anti-CD117-PE (105808), anti-CD117-APC/Cy7 (105825), anti-CD45R-PE/Cy5 (103209), anti-CD45R-APC (103212), anti-Ly--6A/E-APC (108111), anti-Ly-6A/E-Alexa Fluor700 (108142), anti-CD34-PerCP/Cyannine5.5 (128607), anti-CD135-APC (135309), anti-lineage cocktail-Pacific Blue (133305), anti-CD127-PE (121111), anti-CD127-APC(135011), anti-CD11b-FITC (101205), and anti-CD24-Pacific Blue (101819). All these antibodies were purchased from BioLegend. Samples were analyzed using cytometer CytoFlex (Beckman Coulter) and FlowJo software version 10.4. A total of 50,000 events were collected for each sample.

## Preparation of mice serum

For serum collection, mice were anesthetized with isoflurane and blood samples were collected from the ophthalmic vein. Samples were then centrifuged at 5000 rpm for 5 min. Supernatants were transferred to a new tube and centrifuged at 5000 rpm for 5 min again. Supernatants were collected to a new tube and treated with liquid nitrogen fastly and then stored at –80°C.

## Enzyme-linked immunosorbent assay (ELISA)

ELISA was performed as per the manufacturer's instructions (Jianglai). Briefly, working standards and diluted samples were prepared and added to each well. Plates were sealed and incubated for 1 hr at 37°C. After washing three times, 100 μl enzyme-labeled reagents were added and plates were

incubated for 1 hr at 37°C. Finally, TMB substrates were added and incubated for 15–30 min at 37°C followed by Stop solution addition. Then, plates were read at 450 nm within 5 min.

## Singe-cell collection, library construction, and sequencing

Bone marrow cells from WT and DTA[het] mice were flushed and sieved through a 70 μm cell strainer. After red blood cell analysis, dissociated single cells were stained with AO/PI for viability assessment. scRNA-seq was performed using 10X Genomics Chromium platform. Related operations, including generation of gel beads in emulsion (GEMs), barcoding, GEM-RT cleanup, complementary DNA amplification, and library construction, were all carried out following the manufacturer's protocol. By using 150-base-pair paired-end reads, the final libraries were sequenced on the Illumina NovaSeq 6000 platform. The scRNA-seq data could be accessed from GEO database (GSE202516, secure token for reviewer: ihudckqqxvopruz).

## Data processing, dimension reduction, unsupervised clustering, and annotation

ScRNA-seq data analysis was performed by NovelBio Co., Ltd with NovelBrain Cloud Analysis Platform (https://www.novelbrain.com). Fastp was applied with default parameters filtering the adaptor sequence, and the low-quality reads were removed to achieve the clean data. Then, the feature-barcode matrices were obtained by aligning reads to the mouse genome (mm10 Ensemble: version 92) using CellRanger v3.1.0. Down-sample analysis among samples sequenced was applied according to the mapped barcoded reads per cell of each sample and finally the aggregated matrix was achieved. Cells contained over 200 expressed genes, mitochondria UMI rate below 20% passed the cell quality filtering, and mitochondria genes were removed in the expression table.

Seurat package (version 3.1.4; https://satijalab.org/seurat/) was used for cell normalization and regression based on the expression table according to the UMI counts of each sample and percent of mitochondria rate to obtain the scaled data. Principal component analysis (PCA) was constructed based on the scaled data with top 2000 high-variable genes and top 10 principles were used for tSNE construction and UMAP construction. Utilizing graph-based cluster method, the unsupervised cell cluster results based on the PCA top 10 principles were acquired, and the marker genes by FindAllMarkers function with Wilcoxon rank-sum test algorithm were calculated using the following criteria: lnFC > 0.25, p-value<0.05, and min.pct > 0.1. To identify the cell type detailed, the clusters of same cell type were selected for re-tSNE analysis, graph-based clustering, and marker analysis.

## Identification of differential gene expression and gene enrichment analysis

To identify differentially expressed genes among samples, the function FindMarkers with Wilcoxon rank-sum test algorithm was used using the following criteria: lnFC > 0.25, p-value<0.05, and min. pct > 0.1. GO analysis was performed to facilitate elucidating the biological implications of marker genes and differentially expressed genes. The GO annotations from NCBI (http://www.ncbi.nlm.nih.gov/), UniProt (http://www.uniprot.org/), and the Gene Ontology (http://www.geneontology.org/) were downloaded. Fisher's exact test was applied to identify the significant GO categories, and FDR was used to correct the p-values. Pathway analysis was used to find out the significant pathway of the marker genes and differentially expressed genes according to KEGG database. Fisher's exact test was applied to select the significant pathway, and the threshold of significance was defined by p-value and FDR. To characterize the relative activation of a given gene set such as pathway activation, QuSAGE (2.16.1) analysis was performed, and related gene sets involving neutrophil function and SASP were according to the publications (*Xie et al., 2020*; *Zhang et al., 2021*) and are listed in *Supplementary file 2*. Briefly, based on the gene set, the gene set variation analysis (GSVA) software package (*Hänzelmann et al., 2013*) was used to calculate the score of SASP in each cells. Ggpubr R package via the Wilcoxon test (version 0.2.4; https://github.com/kassambara/ggpubr; *Kassambara, 2020*) was used to analyze changes in the scores between WT and DTA[het] mice.

## Developmental trajectory inference and RNA velocity analysis

The single-cell trajectories analysis was applied utilizing Monocle2 (https://cole-trapnell-lab.github.io/monocle-release; *Trapnell, 2019*) using DDR-Tree and default parameter. Before Monocle analysis,

marker genes of the Seurat clustering result and raw expression counts of the cell passed filtering were selected. Based on the pseudotime analysis, branch expression analysis modeling (BEAM Analysis) was applied for branch fate-determined gene analysis. To estimate the cell dynamics, RNA velocity analysis was performed through scVelo package (version 0.2.3) based on ScanPy package (version v1.5.0) with default parameters.

## Statistical analysis

All data were analyzed using GraphPad Prism (v8.2.1) software for statistical significance. p-Value was determined by the Student's *t*-test for two-group or one-way ANOVA test for multiple-group comparisons. Gehan–Breslow–Wilcoxon test was used for analyzing Kaplan–Meier curve of WT and DTA<sup>het</sup> mice.

## Acknowledgements

This work was supported by the National Natural Science Foundation of China (82002339 to JJG, 81820108020 to CQZ) and the Shanghai Frontiers Science Center of Degeneration and Regeneration in Skeletal System (BJ1-9000-22-4002).

## Additional information

### Funding

| Funder | Grant reference number | Author |
| --- | --- | --- |
| National Natural Science Foundation of China | 82002339 | Junjie Gao |
| National Natural Science Foundation of China | 81820108020 | Changqing Zhang |
| Shanghai Frontiers Science Center of Degeneration and Regeneration in Skeletal System | BJ1-9000-22-4002 | Changqing Zhang |

The funders had no role in study design, data collection and interpretation, or the decision to submit the work for publication.

### Author contributions

Peng Ding, Data curation, Software, Formal analysis, Validation, Visualization, Methodology, Writing - original draft; Chuan Gao, Resources, Data curation, Software, Formal analysis, Visualization, Methodology; Youshui Gao, Resources, Data curation, Software, Formal analysis, Validation, Investigation, Visualization, Methodology; Delin Liu, Resources, Data curation; Hao Li, Resources, Data curation, Formal analysis; Jun Xu, Formal analysis, Validation, Methodology; Xiaoyi Chen, Formal analysis, Validation, Visualization; Yigang Huang, Resources, Formal analysis, Validation; Changqing Zhang, Conceptualization, Resources, Supervision, Funding acquisition, Project administration, Writing – review and editing; Minghao Zheng, Conceptualization, Resources, Supervision, Investigation, Visualization, Methodology, Project administration, Writing – review and editing; Junjie Gao, Conceptualization, Resources, Supervision, Funding acquisition, Validation, Investigation, Visualization, Project administration, Writing – review and editing

### Author ORCIDs

Peng Ding http://orcid.org/0000-0002-9348-8134
Junjie Gao http://orcid.org/0000-0003-4820-8524

### Ethics

All mice experiments were approved by the Animal Care and Use Committee of Shanghai Sixth People's Hospital (Permit number: 2021-0935, 2021-0936). All surgery was performed under anesthesia using isoflurane or sodium pentobarbital, and every effort was made to minimize suffering.

Decision letter and Author response
Decision letter https://doi.org/10.7554/eLife.81480.sa1
Author response https://doi.org/10.7554/eLife.81480.sa2

## Additional files

### Supplementary files

• Supplementary file 1. Selected skeleton-related Gene Ontology (GO analysis).

• Supplementary file 2. Gene sets involving neutrophil function and senescence-associated secretory phenotype (SASP).

• MDAR checklist

### Data availability

ScRNA-Seq and RNA-seq data have been deposited into GEO repository with accession codes GSE202516 and GSE202356 respectively. Source data have been deposited in Dryad.

The following datasets were generated:

| Author(s) | Year | Dataset title | Dataset URL | Database and Identifier |
|---|---|---|---|---|
| Ding P, Gao C, Gao Y, Gao J | 2022 | Osteocytes regulate skeletal senescence during development | https://www.ncbi.nlm.nih.gov/geo/query/acc.cgi?acc=GSE202516 | NCBI Gene Expression Omnibus, GSE202516 |
| Ding P, Gao C, Gao Y, Gao J | 2022 | Osteocytes regulate skeletal senescence during development | https://www.ncbi.nlm.nih.gov/geo/query/acc.cgi?acc=GSE202356 | NCBI Gene Expression Omnibus, GSE202356 |
| Ding P, Gao C, Gao Y, Liu D, Li H, Xu J, Chen X, Huang Y, Zhang C, Zheng M, Gao J | 2022 | Data for: Osteocytes regulate senescence of bone and bone marrow | https://dx.doi.org/10.5061/dryad.5tb2rbp6k | Dryad Digital Repository, 10.5061/dryad.5tb2rbp6k |

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
