## [Editor Report]

The work provides a new understanding of the role of osteocytes in regulating other lineage cells in bone, bone marrow, and skeletal muscle. The set of data from the genetic mouse model, bone phenotypic analyses, and scRNA-seq analysis supports the conclusion. This is an important and logically presented study that offers new insight into the biology of osteocytes.

---

## [Decision Letter]

**Decision letter after peer review:**

Thank you for submitting your article "Osteocytes regulate organismal senescence of bone and bone marrow" for consideration by *eLife*. Your article has been reviewed by 3 peer reviewers, including Mei Wan as Reviewing Editor and Reviewer #3, and the evaluation has been overseen by Carlos Isales as the Senior Editor. The following individual involved in review of your submission has agreed to reveal their identity: Jean X Jiang (Reviewer #1);

Essential revisions:

There are several concerns raised by the reviewers that need to be addressed. Particularly, it is necessary for the authors to:

1) Validate the targeting specificity of DMP1-Cre (i.e. preclude muscle targeting) and define how bone partial ablation could be achieved in the mouse model used in the study;

2) Confirm the senescence phenotype in bone/bone marrow and provide an explanation on how SASP score is determined from the scRNA-seq data;

3) Clarify how osteoblast differentiation capacity and osteoclastogenesis were affected in vitro in the absence of osteocytes; and

4) Explore/explain the mechanisms by which osteocyte reduction causes cellular senescence of the surrounding cells. In addition, the authors should consider addressing other concerns raised by the reviewers.

*Reviewer #1 (Recommendations for the authors):*

Quantification and comparison of numbers of skeletal muscle cells between control and DMP-1 DTA will provide a support to exclude the possible activity of DMP1-cre in muscle cells.

*Reviewer #2 (Recommendations for the authors):*

There is a need to correct the reference to the supplemental figures as the numbering started with Figure 2—figure supplemental 1, and this mismatch occurred throughout the manuscript for the remaining figures.

In the generation of mice with reduced osteocyte numbers through conditional expression of DTA in mice in DMP1 expressing osteocytes, it was unclear how partial ablation could be achieved in DTAf/+ mice, as opposed to DTAf/f mice which is lethal at birth. In DTAf/+ mice, toxins are still being produced, although at lower levels throughout the live time of the osteocytes. Does this imply a random process depending on a threshold that could kill the cell? This needs to be better defined and perhaps a need to provide data on the average number of surviving osteocytes with ageing, as an incremental number of osteocytes dying with age could be a reason for the shortened life span.

In the first part of the result section, chronologically, it may be more logical to describe prenatal data first, followed by the 4-week findings.

Line 151. The conclusion that in the context of the mutant mice, the observed osteoporosis, kyphosis, and sarcopenia represent premature ageing needs better justification and to clearly define premature ageing.

While the in vivo data showed changes in osteogenesis and adipogenesis, the in vitro data for impaired osteoblast function is puzzling as these cells are no longer under the influence of osteocytes in this artificial environment. Are the authors proposing a more permanent effect on the osteocytes isolated? If so, what could be the effectors? The same goes for the in vitro differentiation of the bone marrow derived macrophages (Bmms); are there more macrophages primed for the osteoclast lineage? Again, the study was performed in the absence of osteocytes. Clarification and better explanations are needed.

While the relationship between osteoblasts and osteoclasts is tested from an assessment of RANKL/OPG ratio, the relationship between WNT signaling in osteoblasts and osteoprogenitors and the role of SOST produced by osteocytes need to be addressed to provide a fuller picture on the osteogenic lineage.

The scRNA-seq of the 4-week-old bone marrow cells is key to the hypothesis on the changes in haematopoiesis. The data presented on the RNA velocity was not so clear, and importantly, the hypothesis and focus on senescence was not well explained or justified. There is a feel of "cherry picking", and the method of determining a SASP score from the scRNA-seq data was not described, and how was this derived from the data?

Lines 275-276. The conclusion that "osteocyte reduction induced senescence in osteoprogenitors and myeloid lineage cells" could be an over claim as this is at best, an association as there were no supporting functional data. Similarly, the claim for "organismal senescence" is also too strong a statement, as only bone and bone marrow cells were studied.

*Reviewer #3 (Recommendations for the authors):*

1. Introduction- The authors should give a comprehensive summary on the current knowledge about the role of osteocytes in normal bone remodeling as well as in age-associated bone loss.

2. Can the authors comment on whether other organ systems (eg. cardiovascular system, brain…) besides musculoskeletal system have defects in the osteocytes ablation model?

---

## [Author Response]

Essential revisions:There are several concerns raised by the reviewers that need to be addressed. Particularly, it is necessary for the authors to:1) Validate the targeting specificity of DMP1-Cre (i.e. preclude muscle targeting) and define how bone partial ablation could be achieved in the mouse model used in the study;

If *Dmp1*^cre^ directly targets muscle, we would see a reduction of muscle fibers similar to osteocyte ablation in bone. We thus compare number of muscle fibers between WT and DTA^het^ mice at 4 weeks and 13 weeks. As shown, the number of muscle fibers were not altered in these *Dmp1*^cre^ mice at both 4 and 13 weeks *(Lines 165-169, Figure 3 —figure supplement 1A and B)*. In addition, we extracted RNA of muscle and bone from WT mice and performed qPCR to verify the expression of *Dmp1*. qPCR results revealed minimum expression of *Dmp1* in muscle whose cycle threshold was about 29, compared with the expression of *Dmp1* in bone matrix *(Lines 169-173, Figure 3 —figure supplement 1C)*. Together, these data demonstrated that sarcopenia-like phenotype was unlikely caused by *Dmp1*^cre^ expression in muscle cells.

To define how osteocyte partial ablation was achieved, we performed the quantification of empty lacunae ratio of DTA^het^ mice at 13 weeks. About 80% empty lacunae was observed in DTA^het^ mice at 13 weeks which increased about 20% compared to 4 weeks *(Line 126-130, Figure 1 —figure supplement 1B)*, indicating diphtheria toxin (DT) has an accumulative effect with age in DTA^het^ mice. We speculated that when DT accumulated to a threshold, osteocytes were ablated.

2) Confirm the senescence phenotype in bone/bone marrow and provide an explanation on how SASP score is determined from the scRNA-seq data;

SASP score analysis was widely performed to indicate the cellular senescence phenotype (Zhang et al. 2021; Yang et al. 2022; Aging Atlas 2021; Ma et al. 2021; Ma et al. 2020), based on our sc-RNA-seq data, we performed SASP gene set score analysis according to the methods from recent publications (Zhang, et al. 2021) and the gene set has been provided in Supplementary File 2. Briefly, based on the gene set, the gene set variation analysis (GSVA) software package (Hanzelmann, Castelo, and Guinney 2013) was used to calculate the score of SASP in each cells. Ggpubr R package via the Wilcoxin test (https://github.com/kassambara/ggpubr) (version 0.2.4) was used to analyze changes in the scores between WT and DTA^het^ mice. We have updated the detail about SASP score methods in the Materials and methods accordantly *(Lines 717725)*.

To further confirm the senescence phenotype in bone/bone marrow, we performed the senescence β-galactosidase (SA-βGal) staining of frozen sections of WT and DTA^het^ mice femur *(Figure 6 —figure supplement 1D)*. There were obvious SA-βGal^+^ cells in the primary spongiosa, bone marrow and cortical bone in DTA^het^ mice compared to WT mice. Besides, RNA of bone and bone marrow cells from DTA^het^ and WT mice was extracted respectively, followed by qPCR verification. There was significantly increased expression of senescence associated markers including *Cdkn2a*, *Cdkn1a* in DTA^het^ mice bone and bone marrow *(Figure 6 —figure supplement 1B and C) (Lines 308-316)* in DTA mice. Combined with our scRNA-seq data, we are confident to conclude that bone and bone marrow underwent cellular senescence after osteocyte ablation.

3) Clarify how osteoblast differentiation capacity and osteoclastogenesis were affected in vitro in the absence of osteocytes; and

We thank the reviewer for pointing out this issue. We demonstrated that osteoblast and osteoclast activities were affected in vivo after osteocyte ablation, and intriguingly, we found that in vitro differentiation of osteoblast and osteoclast was also affected even in the absence of osteocytes. Previous studies have demonstrated that altered environment can have a long-lasting effects on cells, when these cells are isolated in vitro or even transferred to another mice, they can still perform similar functions or have similar effects based on their memory (Cao et al. 2020; Isaac et al. 2014; Ding et al. 2022; Edgar et al. 2021; Li et al. 2022). Thus, we hypothesized that osteoprogenitors and BMMs have been primed by the altered bone microenvironment in DTA^het^ mice before in vitro differentiation. In our study, bone microenvironment was changed rigorously after osteocyte ablation. After performing the in vitro differentiation of osteogenesis and osteoclastogenesis from different age of mice (4 weeks and 13 weeks) *(Figure 4H-I and Figure 4O-Q)*, we found impaired osteogenesis and promoted osteoclastogenesis, indicating altered bone microenvironment had significant effects on these cells. Importantly, there was a time-dependent accumulative effect on these cells, as when osteoprogenitors and BMMs were collected from older mice, the in vitro osteogenesis and osteoclastogenesis were much more impaired than the cells from younger mice. The underlying mechanism is worth to be further studied. We now have replenished this in Results *(Lines 205-210 and Lines 221-229)* and Discussion *(Lines 391-400)*.

4) Explore/explain the mechanisms by which osteocyte reduction causes cellular senescence of the surrounding cells. In addition, the authors should consider addressing other concerns raised by the reviewers.

We thank the reviewer’s suggestion, and we now have updated this in the Discussion in the revised version *(Line 420-429)*. Different cell lineages harbor in the bone marrow and form specific niches. Each niche has their own specific microenvironment which supports stemness, survival and fates. Also, there are complex cell-cell crosstalk between different niches (Hu et al. 2016). We speculated that osteocytes may regulate cell lineage commitment via releasing factors. As the signaling center, osteocytes have been shown to regulate hematopoiesis, bone remodeling and even remote organs functions (Divieti Pajevic and Krause 2019; Asada, Sato, and Katayama 2015). Our study has provided new finding that osteocyte ablation influences cell lineage commitments and induces senescence. Further study is required to determine factors produced by osteocytes that have impact on lineage commitment and cell senescence.

Reviewer #1 (Recommendations for the authors):Quantification and comparison of numbers of skeletal muscle cells between control and DMP-1 DTA will provide a support to exclude the possible activity of DMP1-cre in muscle cells.

We thank the reviewer’s recommendation, and we quantified and compared the number of skeletal muscle numbers between WT and DTA^het^ mice. Accordingly, the details were given in Response to Essential Revision 1.

Reviewer #2 (Recommendations for the authors):There is a need to correct the reference to the supplemental figures as the numbering started with Figure 2—figure supplemental 1, and this mismatch occurred throughout the manuscript for the remaining figures.

We thank the reviewer for pointing out the issue. In our revised version, we have corrected the numbering of supplement figures which starts from Figure 1 —figure supplement 1, and now the remaining figures were matched with the revised manuscript.

In the generation of mice with reduced osteocyte numbers through conditional expression of DTA in mice in DMP1 expressing osteocytes, it was unclear how partial ablation could be achieved in DTAf/+ mice, as opposed to DTAf/f mice which is lethal at birth. In DTAf/+ mice, toxins are still being produced, although at lower levels throughout the live time of the osteocytes. Does this imply a random process depending on a threshold that could kill the cell? This needs to be better defined and perhaps a need to provide data on the average number of surviving osteocytes with ageing, as an incremental number of osteocytes dying with age could be a reason for the shortened life span.

Please refer to Response to Essential Revision 1.

In the first part of the result section, chronologically, it may be more logical to describe prenatal data first, followed by the 4-week findings.

We thank the useful suggestion from the reviewer. In the revised version, we have changed the order and described prenatal data firstly *(Lines 119-122)*, followed by the 4-week findings.

Line 151. The conclusion that in the context of the mutant mice, the observed osteoporosis, kyphosis, and sarcopenia represent premature ageing needs better justification and to clearly define premature ageing.

We thank the reviewer for pointing out the issue. Premature aging we used in Line 151 means accelerated skeletal aging which occurred at the early stage of the life as opposed to normal aging process which occurred at the late stage of the life (Coppede 2012). As DTA^het^ mice had an obviously shortened lifespan, combined with these age-related phenotypes including osteoporosis, kyphosis and sarcopenia, we concluded that DTA^het^ mice underwent the premature skeletal aging process. We now have changed the ‘premature aging’ into ‘accelerated skeletal aging’ in *Lines 177-179* and clearly defined it.

While the in vivo data showed changes in osteogenesis and adipogenesis, the in vitro data for impaired osteoblast function is puzzling as these cells are no longer under the influence of osteocytes in this artificial environment. Are the authors proposing a more permanent effect on the osteocytes isolated? If so, what could be the effectors? The same goes for the in vitro differentiation of the bone marrow derived macrophages (Bmms); are there more macrophages primed for the osteoclast lineage? Again, the study was performed in the absence of osteocytes. Clarification and better explanations are needed.

We thank the reviewer for pointing out this issue. We have further performed the in vitro osteogenesis and osteoclastogensis obtained from mice of different ages. And we performed flow cytometry analysis and found that osteoclast progenitors were slightly increased *(Lines 219-221) (Figure 4 —figure supplement 1G and H)*. In addition, our sc-RNA seq data demonstrated that osteoclast differentiation related pathway was upregulated in subcluster of monocyte/macrophages *(Figure 6 —figure supplement 1H and I)*. Together, these data indicated that there were more macrophages primed for osteoclast lineage. Accordingly, the details were given in Response to Essential Revision 3.

While the relationship between osteoblasts and osteoclasts is tested from an assessment of RANKL/OPG ratio, the relationship between WNT signaling in osteoblasts and osteoprogenitors and the role of SOST produced by osteocytes need to be addressed to provide a fuller picture on the osteogenic lineage.

According to the reviewer’s suggestion, we further performed GSEA analysis which revealed that osteogenesis related pathways including Wnt signaling pathway, Hedgehog signaling pathway and Notch signaling pathway were downregulated *(Figure 4 —figure supplement 1D, E and F) (Lines 193-196)*. While SOST produced by osteocytes was decreased *(Figure 4 —figure supplement 1B and C)*, we speculated that induction of SASP in osteoprogenitors may be account for the underlying cause. Senescent osteoprogenitors have reduced self-renewal capacity and predominantly differentiate into adipocytes as opposed to osteoblasts (Chen et al. 2016; Li et al. 2017; Rosen et al. 2009). Consistently, our model indicated an increased adipogenesis after osteocyte ablation, and fat-induction factors inhibit osteogenesis during adipogenesis (Chen, et al. 2016). Thus, osteocyte ablation induced senescence accumulation in osteoprogenitors leading to the cell commitment towards adipogenesis with impaired osteogenesis *(Lines 369-391)*.

The scRNA-seq of the 4-week-old bone marrow cells is key to the hypothesis on the changes in haematopoiesis. The data presented on the RNA velocity was not so clear, and importantly, the hypothesis and focus on senescence was not well explained or justified. There is a feel of "cherry picking", and the method of determining a SASP score from the scRNA-seq data was not described, and how was this derived from the data?

Examination of lifespan of DTA mice make us to consider that 4 weeks is the critical time when ablation of osteocyte caused the changes of bone and bone marrow. We thus inferred that bone marrow also underwent significant changes and we further performed scRNA-seq at 4-week-old. RNA velocity is a powerful tool to demonstrate cellular lineage differentiation (Karaman et al. 2022; Bergen et al. 2020; Qiao and Huang 2021). In our study, we found that there was stronger directionality of the velocity vectors in the neutrophil cluster (circled in red) of DTA^het^ mice compared to WT mice *(Figure 5F)*, thus indicating that HSPC differentiation was altered after osteocyte ablation.

As DTA^het^ mice had accelerated skeletal aging phenotypes including myelopoiesis, osteoporosis, kyphosis and sarcopenia with shortened lifespan, and we found that these phenotypes like osteoporosis, sarcopenia, kyphosis and sarcopenia became severe with age. Cellular senescence was the obvious hypothesis that caused acceleration of skeletal aging. As SASP score analysis of scRNA-seq has confirmed our hypothesis, we thus ‘cherry picked’ senescence as our focus and underlying mechanism. We now have replenished it in *Lines 298-302*.

The details about SASP methods were given in Response to Essential Revision 2.

Lines 275-276. The conclusion that "osteocyte reduction induced senescence in osteoprogenitors and myeloid lineage cells" could be an over claim as this is at best, an association as there were no supporting functional data. Similarly, the claim for "organismal senescence" is also too strong a statement, as only bone and bone marrow cells were studied.

We thank the reviewer for pointing out this issue. In the revised version, we adjusted the conclusion we reached which was ‘senescence in osteoprogenitors and myeloid lineage cells was associated with osteocyte reduction’ *(Lines 318-320)*. And we changed the claim for ‘organismal senescence’ into ‘senescence’ instead *(Line 1, Lines 39-40, Lines 105-106, Lines 294-295, Lines 436-437)*.

Reviewer #3 (Recommendations for the authors):1. Introduction- The authors should give a comprehensive summary on the current knowledge about the role of osteocytes in normal bone remodeling as well as in age-associated bone loss.

We thank the reviewer’s useful recommendation, and we now have updated this in the Introduction in the revised version *(Lines 76-92)*.

2. Can the authors comment on whether other organ systems (eg. cardiovascular system, brain…) besides musculoskeletal system have defects in the osteocytes ablation model?

As far as we know, there were some defects in other organ systems like liver, thymus, peripheral white adipose tissue and even brains which have been reported in Dmp1^cre^DTR^fl/fl^ mice (Sato et al. 2013). They found that osteocyte ablation induced lymphoid organ atrophy, thymocyte depletion and altered fat metabolism. Specifically, their data suggested that osteocytes may control fat maintenance in the whole body, including the circulation, and hypothalamus may cooperate it. These data further confirmed the role of osteocyte as an important regulator to other organ systems. We now have also added it in the Discussion *(Lines 429-434)*.

Reference:

Aging Atlas, Consortium. 2021. Aging Atlas: A Multi-Omics Database for Aging Biology. Nucleic Acids Res 49: D825-D830. https://dx.doi.org/10.1093/nar/gkaa894, PMID: 33119753.

Asada, N., M. Sato, and Y. Katayama. 2015. Communication of Bone Cells with Hematopoiesis, Immunity and Energy Metabolism. Bonekey Rep 4: 748. https://dx.doi.org/10.1038/bonekey.2015.117, PMID: 26512322.

Bergen, V., M. Lange, S. Peidli, F. A. Wolf, and F. J. Theis. 2020. Generalizing Rna Velocity to Transient Cell States through Dynamical Modeling. Nat Biotechnol 38: 1408-1414. https://dx.doi.org/10.1038/s41587-020-0591-3, PMID: 32747759.

Cao, H., Q. Yan, D. Wang, Y. Lai, B. Zhou, Q. Zhang, W. Jin, S. Lin, Y. Lei, L. Ma, Y. Guo, Y. Wang, Y. Wang, X. Bai, C. Liu, J. Q. Feng, C. Wu, D. Chen, X. Cao, and G. Xiao. 2020. Focal Adhesion Protein Kindlin-2 Regulates Bone Homeostasis in Mice. Bone Res 8: 2. https://dx.doi.org/10.1038/s41413-0190073-8, PMID: 31934494.

Chen, Q., P. Shou, C. Zheng, M. Jiang, G. Cao, Q. Yang, J. Cao, N. Xie, T. Velletri, X. Zhang, C. Xu, L. Zhang, H. Yang, J. Hou, Y. Wang, and Y. Shi. 2016. Fate Decision of Mesenchymal Stem Cells: Adipocytes or Osteoblasts? Cell Death Differ 23: 1128-39. https://dx.doi.org/10.1038/cdd.2015.168, PMID: 26868907.

Coppede, F. 2012. Premature Aging Syndrome. Adv Exp Med Biol 724: 317-31. https://dx.doi.org/10.1007/978-1-4614-0653-2_24, PMID: 22411253.

Ding, P., Q. Tan, Z. Wei, Q. Chen, C. Wang, L. Qi, L. Wen, C. Zhang, and C. Yao.

2022. Toll-Like Receptor 9 Deficiency Induces Osteoclastic Bone Loss Via Gut Microbiota-Associated Systemic Chronic Inflammation. Bone Res 10: 42.

https://dx.doi.org/10.1038/s41413-022-00210-3, PMID: 35624094.

Divieti Pajevic, P., and D. S. Krause. 2019. Osteocyte Regulation of Bone and Blood. Bone 119: 13-18. https://dx.doi.org/10.1016/j.bone.2018.02.012, PMID: 29458123.

Edgar, L., N. Akbar, A. T. Braithwaite, T. Krausgruber, H. Gallart-Ayala, J. Bailey, A.

L. Corbin, T. E. Khoyratty, J. T. Chai, M. Alkhalil, A. F. Rendeiro, K. Ziberna,

R. Arya, T. J. Cahill, C. Bock, J. Laurencikiene, M. J. Crabtree, M. E.

Lemieux, N. P. Riksen, M. G. Netea, C. E. Wheelock, K. M. Channon, M.

Ryden, I. A. Udalova, R. Carnicer, and R. P. Choudhury. 2021. Hyperglycemia

Induces Trained Immunity in Macrophages and Their Precursors and Promotes Atherosclerosis. Circulation 144: 961-982.

https://dx.doi.org/10.1161/CIRCULATIONAHA.120.046464, PMID: 34255973.

Hanzelmann, S., R. Castelo, and J. Guinney. 2013. Gsva: Gene Set Variation Analysis for Microarray and Rna-Seq Data. BMC Bioinformatics 14: 7.

https://dx.doi.org/10.1186/1471-2105-14-7, PMID: 23323831.

Hu, X., M. Garcia, L. Weng, X. Jung, J. L. Murakami, B. Kumar, C. D. Warden, I. Todorov, and C. C. Chen. 2016. Identification of a Common Mesenchymal Stromal Progenitor for the Adult Haematopoietic Niche. Nat Commun 7:

13095. https://dx.doi.org/10.1038/ncomms13095, PMID: 27721421.

Isaac, J., J. Erthal, J. Gordon, O. Duverger, H. W. Sun, A. C. Lichtler, G. S. Stein, J. B. Lian, and M. I. Morasso. 2014. Dlx3 Regulates Bone Mass by Targeting

Genes Supporting Osteoblast Differentiation and Mineral Homeostasis in vivo. Cell Death Differ 21: 1365-76. https://dx.doi.org/10.1038/cdd.2014.82, PMID: 24948010.

Karaman, S., S. Paavonsalo, K. Heinolainen, M. H. Lackman, A. Ranta, K. A. Hemanthakumar, Y. Kubota, and K. Alitalo. 2022. Interplay of Vascular Endothelial Growth Factor Receptors in Organ-Specific Vessel Maintenance. J Exp Med 219. https://dx.doi.org/10.1084/jem.20210565, PMID: 35050301.

Li, H., P. Liu, S. Xu, Y. Li, J. D. Dekker, B. Li, Y. Fan, Z. Zhang, Y. Hong, G. Yang, T. Tang, Y. Ren, H. O. Tucker, Z. Yao, and X. Guo. 2017. Foxp1 Controls Mesenchymal Stem Cell Commitment and Senescence During Skeletal Aging. J Clin Invest 127: 1241-1253. https://dx.doi.org/10.1172/JCI89511, PMID: 28240601.

Li, X., H. Wang, X. Yu, G. Saha, L. Kalafati, C. Ioannidis, I. Mitroulis, M. G. Netea, T. Chavakis, and G. Hajishengallis. 2022. Maladaptive Innate Immune Training of Myelopoiesis Links Inflammatory Comorbidities. Cell 185: 17091727 e18. https://dx.doi.org/10.1016/j.cell.2022.03.043, PMID: 35483374.

Ma, S., S. Sun, L. Geng, M. Song, W. Wang, Y. Ye, Q. Ji, Z. Zou, S. Wang, X. He, W. Li, C. R. Esteban, X. Long, G. Guo, P. Chan, Q. Zhou, J. C. I. Belmonte, W.

Zhang, J. Qu, and G. H. Liu. 2020. Caloric Restriction Reprograms the SingleCell Transcriptional Landscape of Rattus Norvegicus Aging. Cell 180: 9841001 e22. https://dx.doi.org/10.1016/j.cell.2020.02.008, PMID: 32109414. Ma, S., S. Sun, J. Li, Y. Fan, J. Qu, L. Sun, S. Wang, Y. Zhang, S. Yang, Z. Liu, Z. Wu, S. Zhang, Q. Wang, A. Zheng, S. Duo, Y. Yu, J. C. I. Belmonte, P. Chan, Q. Zhou, M. Song, W. Zhang, and G. H. Liu. 2021. Single-Cell Transcriptomic Atlas of Primate Cardiopulmonary Aging. Cell Res 31: 415-432.

https://dx.doi.org/10.1038/s41422-020-00412-6, PMID: 32913304.

Qiao, C., and Y. Huang. 2021. Representation Learning of Rna Velocity Reveals Robust Cell Transitions. Proc Natl Acad Sci U S A 118.

https://dx.doi.org/10.1073/pnas.2105859118, PMID: 34873054.

Rosen, C. J., C. Ackert-Bicknell, J. P. Rodriguez, and A. M. Pino. 2009. Marrow Fat and the Bone Microenvironment: Developmental, Functional, and

Pathological Implications. Crit Rev Eukaryot Gene Expr 19: 109-24. https://dx.doi.org/10.1615/critreveukargeneexpr.v19.i2.20, PMID: 19392647. Sato, M., N. Asada, Y. Kawano, K. Wakahashi, K. Minagawa, H. Kawano, A. Sada, K. Ikeda, T. Matsui, and Y. Katayama. 2013. Osteocytes Regulate Primary Lymphoid Organs and Fat Metabolism. Cell Metab 18: 749-58.

https://dx.doi.org/10.1016/j.cmet.2013.09.014, PMID: 24140021.

Yang, C., Y. Pang, Y. Huang, F. Ye, X. Chen, Y. Gao, C. Zhang, L. Yao, and J. Gao. 2022. Single-Cell Transcriptomics Identifies Premature Aging Features of Terc-Deficient Mouse Brain and Bone Marrow. Geroscience.

https://dx.doi.org/10.1007/s11357-022-00578-4, PMID: 35545739.

Zhang, H., J. Li, J. Ren, S. Sun, S. Ma, W. Zhang, Y. Yu, Y. Cai, K. Yan, W. Li, B. Hu, P. Chan, G. G. Zhao, J. C. I. Belmonte, Q. Zhou, J. Qu, S. Wang, and G. H.

Liu. 2021. Single-Nucleus Transcriptomic Landscape of Primate Hippocampal Aging. Protein Cell 12: 695-716. https://dx.doi.org/10.1007/s13238-02100852-9, PMID: 34052996.